



# High-resolution Hybrid Inversion of IASI Ammonia Columns to Constrain U.S. Ammonia Emissions Using the CMAQ Adjoint Model

Yilin Chen[1], Huizhong Shen[1], Jennifer Kaiser[1,2], Yongtao Hu[1], Shannon L. Capps[3], Shunliu Zhao[4], Amir Hakami[4], Jhih-Shyang Shih[5], Gertrude K. Pavur[1], Matthew D. Turner[6], Daven K. Henze[7], Jaroslav Resler[8], Athanasios Nenes[9,10], Sergey L. Napelenok[11], Jesse O. Bash[11], Kathleen M. Fahey[11], Gregory R. Carmichael[12], Tianfeng Chai[13], Lieven Clarisse[14], Pierre-François Coheur[14], Martin Van Damme[14], Armistead G. Russell[1]

[1]School of Civil and Environmental Engineering, Georgia Institute of Technology, Atlanta, Georgia 30332, United States

[2]School of Earth and Atmospheric Sciences, Georgia Institute of Technology, Atlanta, Georgia 30332, United States

[3]Department of Civil, Architectural, and Environmental Engineering, Drexel University, Philadelphia, Pennsylvania 19104, United States

[4]Department of Civil and Environmental Engineering, Carleton University, Ottawa, Ontario K1S5B6, Canada

[5]Resources for the Future, Washington, D.C. 20036, USA

[6]SAIC, Stennis Space Center, MS 39529, USA

[7]Mechanical Engineering Department, University of Colorado, Boulder, CO 80309, USA

[8]Institute of Computer Science of the Czech Academy of Sciences, Prague, 182 07, Czech Republic

[9]Institute for Chemical Engineering Sciences, Foundation for Research and Technology Hellas, Patras, GR-26504, Greece

[10]School of Architecture, Civil & Environmental Engineering, Ecole polytechnique fédérale de Lausanne, CH-1015, Lausanne, Switzerland

[11]Atmospheric & Environmental Systems Modeling Division, U.S. EPA, Research Triangle Park, NC 27711, USA

[12]Department of Chemical and Biochemical Engineering, University of Iowa, Iowa City, IA 52242, USA

[13]NOAA Air Resources Laboratory (ARL), Cooperative Institute for Satellites Earth System Studies (CISESS), University of Maryland, College Park, MD 20740, USA

[14]Université libre de Bruxelles (ULB), Spectroscopy, Quantum Chemistry and Atmospheric Remote Sensing (SQUARES), Brussels, Belgium

*Correspondence to:* Armistead G. Russell (ar70@ce.gatech.edu)





**Abstract**

Ammonia ($NH_3$) emissions have large impacts on air quality and nitrogen deposition, influencing human health and the well-being of sensitive ecosystems. Large uncertainties exist in the "bottom-up" $NH_3$ emission inventories due to limited source information and a historical lack of measurements, hindering the assessment of $NH_3$-related environmental impacts. The increasing capability of satellites to measure $NH_3$ abundance and the development of modeling tools enable us to better constrain $NH_3$ emission estimates at high spatial resolution. In this study, we constrain the $NH_3$ emission estimates from the widely used national emission inventory for 2011 (2011 NEI) in the U. S. using Infrared Atmospheric Sounding Interferometer $NH_3$ column density measurements (IASI-$NH_3$) gridded at a 36 km by 36 km horizontal resolution. With a hybrid inverse modeling approach, we use CMAQ and its multiphase adjoint model to optimize $NH_3$ emission estimates in April, July, and October. Our optimized emission estimates suggest that the total $NH_3$ emissions are biased low by 32% in 2011 NEI in April with overestimation in Midwest and underestimation in the Southern States. In July and October, the estimates from NEI agree well with the optimized emission estimates, despite a low bias in hotspot regions. Evaluation of the inversion performance using independent observations shows reduced underestimation in simulated ambient $NH_3$ concentration in all three months and reduced underestimation in $NH_4^+$ wet deposition in April. Implementing the optimized $NH_3$ emission estimates improves the model performance in simulating $PM_{2.5}$ concentration in the Midwest in April. The model results suggest that the estimated contribution of ammonium nitrate would be biased high in NEI-based assessments. The higher emission estimates in this study also imply a higher ecological impact of nitrogen deposition originating from $NH_3$ emissions.

## 1. Introduction

Ammonia ($NH_3$) emissions play a major role in ambient aerosol formation and reactive nitrogen deposition (Stevens, 2019: Houlton et al., 2013). However, our understanding of $NH_3$ sources and sinks is limited by the large uncertainties present in the $NH_3$ emissions inventories (Xu et al., 2019; McQuilling and Adams, 2015). In chemical transport models, uncertainties in $NH_3$ emissions propagate into the dynamic modeling of the atmospheric transport, chemistry, and deposition of $NH_3$, other reactive nitrogen species, and other key atmospheric constituents associated with $NH_3$ (Heald et al., 2012; Paulot et al., 2013; Kelly et al., 2014; Zhang et al., 2018b), hindering an accurate assessment of the various $NH_3$-related environmental impacts and the associated sources. The large uncertainties in the $NH_3$ emission inventories are partially due to a lack of sufficient in-situ $NH_3$ measurements that could be used to constrain emission estimates (Zhu et al., 2015). This work utilizes satellite observations from the Infrared Atmospheric Sounding Interferometer $NH_3$ column density measurements (IASI-$NH_3$) (Clarisse et al., 2009;Van Damme et al., 2017), to provide a high-resolution, optimized $NH_3$ emission inventory for the U.S. developed using an adjoint inverse modeling technique (Li et al., 2019), the robustness of which is demonstrated by evaluation against multiple independent in-situ measurements.





Emerging satellite observations of gaseous $NH_3$ provide a unique opportunity to better constrain the bottom-up $NH_3$ emission estimates for both their spatial distribution and seasonality. Bottom-up inventories calculate the $NH_3$ emissions based on estimated activity levels and corresponding emission factors, both of which are subject to high uncertainties, particularly for agricultural sources, the major contributor (Cooter et al., 2012; McQuilling and Adams, 2015). Several studies have utilized $NH_3$ column density retrieved from IASI (Clarisse et al., 2009; Van Damme et al., 2015b) or the Atmospheric Infrared Sounder (AIRS; (Warner et al., 2016)) as well as the inferred surface mixing ratio of $NH_3$ from the Cross-track Infrared Sounder (CrIS; (Shephard and Cady-Pereira, 2015; Shephard et al., 2019)) to characterize the spatiotemporal distribution of $NH_3$. These satellite measurements are useful for supplementing emission inventories to identify and quantify underestimated or missing emission hotspots, especially in intensive agricultural zones (Van Damme et al., 2018; Dammers et al., 2019; Clarisse et al., 2019). These studies find that the satellite-derived emission estimates are often twice as much as the bottom-up estimates on a regional scale and can be over 10 times higher over hotspots. However, the $NH_3$ retrievals from satellites are also subject to large uncertainties when the signal-to-noise ratio is low, which limits their ability to accurately measure $NH_3$ columns in low emission areas (Clarisse et al., 2010; Van Damme et al., 2015a).

Inverse modeling-based optimization combines the information from *a priori* emission inventories and observations and allows us to use the information from both. As one of the inverse modeling methods, the four-dimensional variational assimilation (4D-Var) method seeks the best emission estimate by minimizing a cost function that measures the differences between observations and model predictions, as well as the differences between a prior and adjusted emission estimates. 4D-Var can be computationally expensive at fine model resolutions or with a large set of observations to be assimilated (Brasseur and Jacob, 2017). Recent studies took advantage of the implementation of the adjoint technique in the chemical transport models to conduct 4D-Var for optimizing emissions estimation (Zhu et al., 2013; Paulot et al., 2014; Zhang et al., 2018c). The adjoint-based inversion method calculates the gradients of a cost function analytically and searches the solution using a steepest-descent optimization algorithm through iterating (Brasseur and Jacob, 2017). By testing the performance of the inverse modeling method using artificial observational data, Li et al. (2019) proposed that a two-step optimization process, which combines the iterative mass balance (IMB) method and the 4D-Var method, can further reduce the computational cost. The IMB method assumes a linear relationship between the $NH_3$ column density and local $NH_3$ emission and searches the emission scaling factors iteratively until the simulated $NH_3$ column density converges to the observations. At a coarse (2°×2.5°) resolution, the IMB method is as effective as the 4D-Var method and requires 2/3 less computational time. In the second step, emission scaling factors obtained from the IMB method with a coarser resolution are used as an initial starting point for 4D-Var optimization process to reduce the overall computational time (Li et al., 2019).

In this study, the IASI-$NH_3$ dataset was applied to optimize $NH_3$ emission estimates from the 2011 National Emission Inventory (NEI 2011) using CMAQ and its adjoint model at a 36 km×36 km resolution. The multiphase adjoint model for CMAQ v5.0 was developed recently, including full adjoints for gas-phase chemistry, aerosols, cloud process, diffusion, and advection (Zhao et al., 2019). Both process-by-process and full adjoint model



evaluations show reasonable accuracy based on agreements between the adjoint sensitivities and forward sensitivities (Zhao et al., 2019). Previous inversion based NH$_3$ emission constraint using in-situ measures are limited by the spatial coverage and representativeness of the measurements (Gilliland et al., 2006; Henze et al., 2009; Paulot et al., 2014;). Zhu et al. (2013) first attempted to optimize NH$_3$ emission inventory using NH$_3$ derived from the Tropospheric Emission Spectrometer (TES) satellite at 2°×2.5° resolution (Zhu et al., 2013). Inverse modeling at such a coarse resolution is limited to refining regional emissions. Similar to the inversion using CrIS NH3 measurements (Cao et al., 2020), inversion with the IASI-NH$_3$ dataset allows us to perform the optimization at a finer resolution with its daily global spatial coverage. Besides, the hybrid inversion approach adopted in this study allows us to calculate full adjoint sensitivities online instead of using approximated sensitivities from the offline-simulations (Zhu et al., 2013, Cao et al., 2020). The performance of our optimized estimates and the NEI 2011 are evaluated and compared based on in-situ observed ambient NH$_3$ concentrations and NH$_4^+$ wet deposition. Finally, by substituting the *a priori* NH$_3$ emissions with the optimized emissions, we assess the subsequent changes in simulated ambient PM$_{2.5}$ concentrations and nitrogen deposition exceedances.

## 2. Materials and Methods

### 2.1 IASI-NH$_3$ observations

NH$_3$ column densities retrieved from IASI onboard the Metop-A satellite are assimilated to constrain spatially-resolved NH$_3$ emissions using the 2011 NEI as the *a priori* inventory (Clarisse et al., 2009; Van Damme et al., 2014; USEPA, 2014). The polar sun-synchronous satellite has a 12-km diameter footprint at nadir and a bidaily global coverage. Only observations from the morning pass around 9:30 am local standard time (LST) are used due to more favorable thermal contrast and smaller errors comparing to the ones from the night pass around 9:30 pm (LST). A comparison between the IASI-NH$_3$ data and ground-based Fourier transform infrared observations shows a correlation between the two with r = 0.8 and the slope = 0.73, indicating a tendency of IASI-NH$_3$ to underestimate the FTIR observations (Dammers et al., 2016). A comparison between IASI-NH$_3$ and airborne measurements also indicated an underestimation in California, while the comparison between IASI-NH$_3$ and ground observation from Ammonia Monitoring Network (AMoN) network indicated an overestimation (Van Damme et al., 2015a; NADP, 2014). Overall, the evaluations show broad consistency between IASI-NH$_3$ and other independent measurements with no consistent biases identified. These evaluations were based on previous datasets. Here we use a new version that relies on another retrieval algorithm, which among other things has a better performance for measurements under unfavorable conditions (Whitburn et al., 2016; Van Damme et al., 2017).

Specifically, the NH$_3$ products for 2011 from ANNI-NH$_3$-v2.2R-I datasets were used (Van Damme et al., 2017). The algorithm relies on the conversion of hyperspectral range indices to NH$_3$ column density using a neural network that takes into account 20 input parameters characterizing temperature, pressure, humidity, and NH$_3$ vertical profiles. A relative uncertainty estimate is provided along with each of the NH$_3$ vertical column density in the dataset. Small negative columns are possible – and these are valid observations, needed to reduce overall biases in the dataset. As





the retrieval is unconstrained, no averaging kernels are calculated. We therefore directly compare the IASI-NH$_3$ column density with the simulated column density in CMAQ. Such comparison may be biased because the sensitivity of retrieved NH$_3$ column densities to NH$_3$ concentrations is height-dependent (typically peaks around 700 – 850 hPa) (Dammers et al., 2017; Shephard et al., 2015). Although the CMAQ simulated NH$_3$ columns are also

most sensitive to NH$_3$ concentration changes between 700 to 900 hPa (**Figure S1**), we cannot quantify the relating uncertainties without knowing the averaging kernels. Without information on averaging kernels, differences between NH$_3$ vertical profiles in CMAQ and the ones used for retrieval may also contribute to the bias between retrieved and modeled column densities, depending on the magnitude of differences (Whitburn et al., 2016).

The NH$_3$ retrieved columns densities are regridded to the 36-km by 36-km CMAQ grid for 4D-Var data

assimilation, and 216-km by 216-km resolution (a 6 grid by 6 grid CMAQ simulation grid matrix) for iterative mass balance (**Figure 1**). The mean column density ($\Omega_o$) is calculated as the monthly arithmetic mean of all retrievals with their centroids falling in the same grid, following the recommendation that the unweighted mean is preferred for the updated version of IASI-NH$_3$ as error-weighting can lead to biases (Van Damme et al., 2017). The relative error (molec/cm$^2$) corresponding to mean column density in each grid is calculated following Van Damme et al.

(2014) as:

$$\bar{\sigma} = \frac{\Sigma \frac{1}{\sigma_i}}{\Sigma \frac{1}{\sigma_i^2}} \times \Omega_o \tag{1}$$

where $\bar{\sigma}$ is mean relative error (molec/cm$^2$), $\sigma_i$ is the relative error associated with each NH$_3$ column density retrieval as reported, and $\Omega_o$ is the mean column density (Van Damme et al., 2014).

The observations from April, July, and October are used to constrain the monthly NH$_3$ emission estimates in

corresponding months from 2011 NEI. Observations from winter months are not used because they are too noisy when the thermal contrast is low (Dammers et al., 2016).

**2.2 NH$_3$ emission from 2011 NEI**

The EPA 2011 NEI is used as *a priori* emission estimates. Major NH$_3$ sources include livestock waste management, fertilizer application, mobile sources, fire, and fuel combustion, with the majority being emitted by the first two

sources. Specifically, the emissions from livestock waste management are estimated based on county-level animal population data and process-based daily emission factors. Emissions from fertilizer applications are estimated based on county-level fertilizer quantities and fixed emission factors, following the CMU ammonia Model (USEPA, 2015). The NH$_3$ emissions over Mexico and Canada are derived from the simulation results of a fully coupled bi-directional agroecosystem and chemical-transport model (FEST_C_EPIC_CMAQ_BIDI) (Shen et al., 2019).

Emissions for other species also come from the 2011 NEI.

**2.3 CMAQ and its adjoint**



We use the Community Multiscale Air Quality Modeling System (CMAQ) v5.0 (Byun and Schere, 2006; USEPA, 2012) and its adjoint (Zhao et al., 2019), driven by meteorological fields produced from the Weather Research and Forecasting (WRF) Model v3.8.1 with grid nudging using the North American Regional Reanalysis (NARR) dataset
(NOAA, 2019). The CB05 chemical mechanism was adopted for gas-phase chemistry (Yarwood et al., 2005). The model implements ISORROPIA-II in the aerosol module (AERO06) to calculate the gas-particle partitioning of $NH_3$ and $NH_4^+$ (Fountoukis and Nenes, 2007). The simulation domain covers the contiguous U.S. (CONUS) and part of Canada and Mexico with a 36 km by 36 km horizontal resolution and 13 vertical layers extending up to 14.5 KPa (~16 km) (**Figure 1**). Monthly simulations are conducted for April, July, and October in 2011 with a 10-day spin-up
for each month.

**2.4 Hybrid inversion approach**

We chose the hybrid inversion approach to combine the advantage of the faster computational speed of the mass balance method and the better optimization performance of the 4D-Var method. The first step is to apply the IMB approach to adjust the *a priori* (2011 NEI) $NH_3$ emission at 216 km by 216 km resolution (referred as the coarse grid
hereafter) based on the ratio between the monthly-averaged observed ($\Omega_o$) and simulated ($\Omega_a$) $NH_3$ column density at the satellite overpassing time, iteratively. At each iteration, the emission in each grid is scaled by the ratio following the equation below,

$$E_t = \frac{\Omega_o}{\Omega_a} \times E_a \qquad (2)$$

where $E_t$ and $E_a$ are the new and *a priori* emission estimates, respectively. The method has been described in detail
in previous studies (Li et al., 2019; Cooper et al., 2017; Martin et al., 2003). The IMB is applied at the coarse grid so that the $NH_3$ column will be dominated by the local emissions instead of transport from neighboring grids (Li et al., 2019). The coarse resolution also reduces the uncertainty associated with IASI-$NH_3$ as the number of retrievals increases in each grid. For grids with mean relative error larger than 100%, the satellite observations are considered to be too noisy to provide useful constraints and the *a priori* emission estimates are retained. The iteration stops
when the normalized mean square error either decreases by less than 10% or begins to increase. The final scaling factor ($\varepsilon_0$) for each grid is the multiplication of the scaling factors derived at each iteration and downscaled to 36 km by 36 km resolution by assigning the same value to the 6 by 6 grid matrix. This scaling factor is applied to the 2011 NEI emissions to create the revised *a priori* estimate for the 4D-Var inversion.

Next, the 4D-Var inversion is performed. The solution of the optimization problem is sought iteratively by
minimizing the cost function (J) defined as the combination of error-weighted, squared difference between emission scaling factor and unity and the error weighted, squared difference between IASI-$NH_3$ and the simulated column density, as below:

$$J = \gamma(\varepsilon - \varepsilon_0)^T S_a^{-1}(\varepsilon - \varepsilon_0) + (\Omega_o - F(\varepsilon))^T S_o^{-1}(\Omega_o - F(\varepsilon)) \qquad (3)$$



$\varepsilon$ is the monthly emission scaling factor to be optimized at each iteration where $\varepsilon = \log\left(\frac{E_t}{E_a}\right)$ on the 36 km by 36 km CMAQ grid, consisting of 6104 elements overland grid cells in CONUS. $S_a$ and $S_o$ are error covariance matrices for the *a priori* emission estimates and IASI-NH$_3$ retrievals, respectively. The two matrices are assumed to be diagonal. For $S_o$, the grid average absolute error is used to represent the observational error. To reduce the influence of retrievals close to or below the detection limit, an estimated detection limit of $4.8\times10^{15}$ molecules/cm$^2$ is added to the $S_o$ (Dammers et al., 2019). Our test shows that negative $\Omega_o$ will lead to a continuous decrease in the adjusted emission for the grid cell because modeled column density cannot become negative. To limit the influence of these negative $\Omega_o$, their original weights are multiplied by 0.01. For $S_a$, the uncertainty in each grid is assumed to be 100% of the *a priori* emissions. F($\varepsilon$) is CMAQ simulated NH$_3$ column density sampled at the satellite passing time if there is at least one IASI-NH$_3$ retrieval in that grid; $\gamma$ is the regularization factor balancing the relative contribution of the *a priori* emission inventory and IASI-NH$_3$ retrievals to the J value. $\gamma$ is chosen to be 30 for all 3 months based on the L-curve criteria (Hansen, 1999) (**Figure S2**).

The gradients of the cost function to NH$_3$ emissions are calculated by the CMAQ adjoint model. In each iteration, the emission-weighted monthly averaged sensitivities in each grid are supplied to the L-BFGS-B optimization routine contained in the "optimr" package in R to find the scaling factors that will achieve the minimum of the cost function (Zhu et al., 1997; Byrd et al., 1995). NH$_3$ column density is re-simulated using adjusted emissions by the new set of scaling factors. The iteration process is terminated when the decrease in J is less than 2% or the local minimum is reached.

### 2.5 Posterior evaluation

The posterior emissions are evaluated by comparing the model simulation from optimized emissions with observations. Simulated results are compared with ambient NH$_3$ concentrations from the AMoN (NADP, 2014), and the NH$_4^+$ wet deposition from the National Atmospheric Deposition Program (NADP, 2019). The simulated NH$_3$ concentration in ppmV is converted to µg/m$^3$ using local temperature and pressure from the model meteorological inputs. For evaluation against the NH$_4^+$ wet deposition, the simulated deposition is scaled by the ratio between measured and simulated precipitation to eliminate the bias introduced by precipitation fields (Appel et al., 2011).

### 3. Results

### 3.1 Optimization performance evaluation

The optimized NH$_3$ emissions reduce the bias in the NH$_3$ columns between the satellite observation and the model prediction as shown by the decrease in the values of normalized root mean square error (NRMSE) and normalized mean biases (NMBs) in **Figure 2**. There are negative biases using 2011 NEI in all three months, especially in areas with high emission rates. Although the IMB inversion can lower the NRMSE, it tends to over-adjust and introduce a positive bias likely because of the coarse resolution and neglect of the impact of transport. The 4D-Var inversion



effectively decreases the positive bias and further reduces the NRMSE. The cost function value reduces by 50%, 57%, and 34% with the 4D-Var inversion in April, July, and October, respectively. We find that it is more challenging to adjust the emissions in April than in the other two months because of the greater differences in the magnitude and the spatial distribution of the emissions. The optimized $NH_3$ emission successfully captures the high

$NH_3$ column density in the Southern States, reducing the NRMSE by 98%. Despite the general improvement in the model performance, negative biases in July increase in California's San Joaquin Valley. Scaling up the emission in the San Joaquin Valley will result in high $NH_3$ concentrations downwind even when the local $NH_3$ emissions downwind are zeroed, whereas the IASI-$NH_3$ observed concentrations downwind are low. The transported hotspot downwind of the San Joaquin Valley in CMAQ only occurs in July, suggesting near field removal may not be

captured at the current resolution, and warrants further investigation. Grid by grid comparison between model-simulated $NH_3$ column density using the *a priori* and optimized estimates with IASI-$NH_3$ shows improved agreement in both high and low emission grids after optimization (**Figure S3**). It shows that the hybrid inversion approach can alleviate the weakness of direct 4D-Var inversion which tends to over-adjust high emission regions and under-adjust low emission regions, mainly because the IMB inversion provides a better initial state.

The IMB inversion took three iterations to achieve the convergence condition for each month, and subsequently, the 4D-Var inversion took five, four, and three iterations for April, July, and October, respectively. Fewer iterations are needed with the hybrid approach than the direct 4D-Var inversion which typically takes up to 15 to 20 iterations of adjoint simulation (Paulot et al., 2014; Zhang et al., 2018a). The CPU time of a forward simulation is only 1/5 of an adjoint simulation. In total, the CPU time required by the hybrid approach is expected to be 60% lower than the

direct 4D-Var inversion approach.

### 3.2 Optimized estimate of $NH_3$ emissions

The monthly total $NH_3$ emission in CONUS increases by 46% in April, 6.6% in July, and 6.9% in October for the optimized estimates, respectively. Spatially, the distribution for high emission regions shifts from Midwest in the 2011 NEI to the Southern States in the optimized estimates in April, whereas the hot spot regions remain consistent in July and October (**Figure 3**). Regional total emissions are summarized according to the USDA Farm Production

regions, which defines the areas with similar crop production activities (Cooter et al., 2012). In general, the regional variation of $NH_3$ emissions in April is dominated by fertilizer application. The optimized estimates in regions with high contributions from fertilizer applications in 2011 NEI, including the Corn Belt, Lake States, and Northern Plains, are lower than the 2011 NEI. In contrast, the optimized estimates are 2 – 3 times higher than the 2011 NEI

estimates in the Delta States and Southern States where the *a priori* estimates for $NH_3$ emission from fertilizer application are low. The optimized $NH_3$ emission pattern in April is more consistent with the spatial pattern of inorganic nitrogen fertilizer estimated based on plant demand (Cooter et al., 2012) as well as the livestock population distribution (USDA, 2012), suggesting the potential bias in the agricultural practices used in 2011 NEI. In July, regional differences are smaller except for the Northern Plain and Southeast. In the Northern Plain, the $NH_3$

emission is 66% higher in the optimized estimates, driven by the emission increase in hotspot areas with



concentrated animal feeding operations (CAFO) (USDA, 2012; Van Damme et al., 2017, Clarisse et al., 2019). The potential bias in different sectors suggests the need for sectoral inversion when a larger observational dataset becomes available in the future. In the Southeast, the IASI-NH$_3$ column densities are very low, even over known CAFO sites, and had high errors associated with the retrievals because of the low thermal contrast and a smaller

number of retrievals (Schiferl et al., 2014). The negative increment in the Pacific region is due to the disagreement between modeled high NH$_3$ columns and observed low values from IASI-NH$_3$ downwind of the San Joaquin Valley of California, as discussed previously. In October, the relative difference is less than 10% in most of the regions, indicating that the 2011 NEI appropriately reflects the NH$_3$ emission pattern. There is a significant increase in the NH$_3$ emissions in Mexico during all three months. Such an emission increment is crucial to improving the model

performance in both Mexico and the southwestern U. S. However, it was not a goal of this study to determine emissions biases in Mexico given the limited information on NH$_3$ emissions.

The total NH$_3$ emissions in the optimized estimates are 671 Gg, 500 Gg, and 320 Gg per month in April, July, and October, respectively. Similar to a bottom-up agricultural NH$_3$ emission inventory (MASAGE_NH$_3$) and two inverse model optimized estimates based on NH$_4^+$ wet deposition, we find a higher emission in the spring season

(Paulot et al., 2014; Gilliland et al., 2006), while others, including the NEI, estimates a summertime peak (Zhu et al., 2013; USEPA, 2015; Cooter et al., 2012). The large variation between different inventories warrants both improved information on bottom-up inventories and more observations to support inverse model optimization in the spring season. Better knowledge about agricultural activities and more independent ground and space observations are needed. It should also be noted that there are interannual variations in emission inventories developed for different

years. The total emission estimates in July and October are closer to the 2011 NEI estimates than those estimates from other emission inventories and inverse analysis. The good agreement with IASI-NH$_3$ indicates that the 2011 NEI captures the NH$_3$ emission pattern in general in these two months.

**3.3 Evaluation of the optimized emission estimates against independent datasets**

The robustness of the NH$_3$ emission optimization is evaluated by comparing the model outputs using both the *a*

*priori* and optimized emission estimates with independent observations. The bias and uncertainties inherited in the CMAQ forward model and its adjoint, as well as the assumptions made about the uncertainties of the *a priori* emission inventory and IASI-NH$_3$ observations, will all influence the robustness. Here, we choose to evaluate the outputs against 1) biweekly average ambient NH$_3$ concentrations measured by AMoN; 2) weekly average NH$_4^+$ wet deposition measured by NADP (**Figure 4**).

In general, the optimized NH$_3$ emission reduces the negative NMB when comparing the CMAQ outputs with AMoN NH$_3$ concentration for all three months. Yet, the NRMSE gets higher and R$^2$ gets lower, indicating a higher spatial variation in the residuals. This is likely due to the tendency of satellite-based inversion to over-adjust emissions in high concentration areas (Zhu et al., 2013). There is a greater improvement at the high concentration end than the low concentration end because both IASI satellite and the passive samplers at the AMoN sites have higher



uncertainties in areas with low NH$_3$ abundance (Van Damme et al., 2015a; Puchalski et al., 2011). There is an over-adjustment for sites in Pennsylvania in April where there is a hotspot observed by IASI. The hotspot in monthly average is dominated by high NH$_3$ column densities observed in April 14$^{th}$ and 15$^{th}$, possibly from a large transported plume from the central U.S. to Pennsylvania (**Figure S4**). The fact that it is transported at higher altitude in 2 days could explain that it is not measured by ground observations at AMoN sites at biweekly resolution. The

long-range transport at higher altitude may lead to an overestimation in IASI retrieved NH$_3$ column densities which assume a vertical profile with highest concentrations near the surface (Whitburn et al., 2016). Such transport is also not well represented in the hybrid inverse modeling approach where the transport effect is not included in the IMB inversion at 216 km by 216 km resolution.

For evaluation against NADP observations, there is a noticeably improved agreement in April with reduced negative

NMB and reduced discrepancies for most of the data pairs. For July, the emission optimization only slightly improved the model performance. For October, the optimization increased the NMB from -1.8% to 10%. It indicates that NH$_3$ emission is not the dominant explanatory factor for bias in simulated NH$_4^+$ wet deposition that is commonly observed in chemical transport models (Appel et al., 2011; Paulot et al., 2014). Overall, the improved model operational performance for ambient NH$_3$ suggests that the inverse model optimization applied in this study

provides improvements in the NH$_3$ emission estimates during all three months in most of the CONUS.

## 4. Implications

### 4.1 Ambient aerosol concentration

As a major precursor of ambient aerosol formation, the NH$_3$ emission inventory is believed to be a major source of uncertainty in PM$_{2.5}$ assessment in several parts of the CONUS (Henze et al., 2009; Schiferl et al., 2014; Heald et

al., 2012), which can further bias the source contribution assessments on PM$_{2.5}$-related health impacts (Lee et al., 2015, Zhao et al., 2019). Comparison of the simulated PM$_{2.5}$ mass concentration using the *a priori* and optimized NH$_3$ emission estimates shows that the NH$_3$ emission bias in April is a major factor for bias in the modeled PM$_{2.5}$ concentration leading to high or low bias in ammonium nitrate (NH$_4$NO$_3$) formation (**Figure 5**). The relative change of the monthly average PM$_{2.5}$ concentration is over 10% in one-fifth of the CONUS, including an increase in the

Northeastern, Pacific West, and Rocky Mountains regions, and a decrease in the Midwest. For most of these regions, over 90% of the change is driven by the change in concentration of NH$_4^+$ and NO$_3^-$.

Comparison of the simulated monthly average NH$_4^+$ and NO$_3^-$ concentration using the *a priori* estimates against ambient monitoring network data (USEPA, 2018) shows that there is a high bias in the Midwest region and Pennsylvania state, and underestimation low bias for the rest of the sites (**Table 1**). Simulations using the optimized

NH$_3$ emission estimates reduce the high bias in the Midwest region but exacerbate the high bias in the Pennsylvania state and surrounding areas. For the other sites, the impact of optimization is mixed but minor in general.



For the Midwest, our optimized $NH_3$ emission is 31% lower than the 2011 NEI, leading to a 20 - 30% decrease in $NH_4^+$ and $NO_3^-$ concentration. Overestimation of $NO_3^-$ in the Midwest has been recognized in previous model evaluations. Previous studies attempted to moderate the high bias by lowering the nitric acid ($HNO_3$) concentration

through either lowering both daytime and nighttime $HNO_3$ formation rate or raising deposition removal rate (Heald et al., 2012; Zhang et al., 2012; Walker et al., 2012). It was found that such modification in the model parameterization cannot fully account for the overestimation (Heald et al., 2012; Zhang et al., 2012; Walker et al., 2012). Our study implies that the springtime overestimation can partly be explained by the overestimation in $NH_3$ emissions which drives the high bias in $NH_4NO_3$ formation.

The large increase of the $NH_4NO_3$ concentration in Pennsylvania state and surrounding areas is due to the over-amplified local $NH_3$ emissions in the optimized estimates to match the high $NH_3$ column density in IASI-$NH_3$ 2011, as discussed earlier. It adds to the existing overestimation in $NH_4^+$ and $NO_3^-$ concentration as compared to ground measurements. The fact that the simulated ambient $NH_3$ concentration, $NH_4^+$ concentration, and $NH_4^+$ wet deposition using the optimized $NH_3$ estimates is biased high in comparison with independent ground measurements

suggests the enhanced $NH_3$ abundance observed from IASI is driven by long-range transport at higher altitudes instead of local surface emissions.

For the rest of the CONUS, there is only a slight impact of the optimization on simulated $NH_4NO_3$ formation. For example, although the $NH_3$ emission is doubled in the San Joaquin Valley in California, the modeled $NH_4^+$ and $NO_3^-$ concentrations are still biased low using the optimized estimates. A sensitivity test using GEOS-Chem shows that

the San Joaquin Valley region is nitric acid-limited instead of ammonia-limited (Walker et al., 2012), suggesting that there is an underestimation in $HNO_3$ formation. A comparison of the simulated and measured speciated $PM_{2.5}$ shows that there is a low bias in non-volatile cation concentrations in the sites in the San Joaquin Valley, limiting the formation of $NH_4NO_3$ through gas-particle partitioning (Chen et al., 2019). Thus, attempts to close the gap between the simulated and monitored $NH_4^+$ and $NO_3^-$ concentrations by scaling $NH_3$ emission alone are ineffective and might

lead to an overestimation in local $NH_3$ emissions.

For July and October, there is a very limited difference between the simulated $PM_{2.5}$ concentration using the optimized and *a priori* $NH_3$ emission estimates, as expected, because the change in $NH_3$ emission is small. There are only 1% and 4% of the CONUS with a relative change in $PM_{2.5}$ concentration over 10%. This result shows that the uncertainty in $NH_3$ emission estimates is moderate and is not a major contributor to biases in modeled $PM_{2.5}$ in July

and October.

## 4.2 Reactive nitrogen (Nr) deposition

The uncertainties in $NH_3$ emission inventory also impact the reactive nitrogen (Nr) deposition assessment, which informs the ecosystem impacts evaluation and effective mitigation actions (Ellis et al., 2013). To evaluate the impact of the $NH_3$ emission optimization on simulated Nr deposition, the Nr deposition amount simulated using optimized

and *a priori* emission estimates is analyzed in all biodiversity-protected areas designated by the USGS (**Figure S5**)





within CONUS (USGS, 2018). In total, the Nr deposition increased by 39%, 2%, and 9% on average in these protected areas in April, July, and October, respectively. A regional comparison based on the Level I ecoregions (Pardo et al., 2015) shows that the deposition increment is the highest in the Great Plain region (+73%), followed by the Eastern Temperate Forest (+41%) (**Figure 6**). Although the overall increase is small in July and October, the increment can be high in individual ecoregions, including Southern Semiarid Highlands (+109% in July), Temperate Sierras (+66% in July), and Marine West Coast (+64% in October). In addition to the increment in deposition amount, higher $NH_3$ emission, especially in intensive agriculture regions, may indicate higher source contribution from agricultural $NH_3$ than previous estimates (Lee et al., 2016).

Driven by the increase in the reduced form of Nr ($NH_3$ and $NH_4^+$) deposition, a higher share of reduced form of Nr to the total Nr deposition is found in most of the ecoregions for all three months than the NEI-based estimates. More detrimental impacts on sensitive species and biodiversity are expected when this change in dominant Nr form are considered in addition to the increase in magnitude because the growth of many sensitive plant species will be inhibited by a high $NH_4^+$ to $NO_3^-$ ratio in soil and water (Bobbink and Hicks, 2014).

## 5. Conclusions

We apply the newly developed multiphase adjoint of the CMAQ v5.0 chemical transport model and $NH_3$ column observations from the satellite-borne IASI to optimize $NH_3$ emissions estimates in the CONUS using a hybrid inversion modeling approach. The approach consists of a coarse-resolution iterative mass balance inversion (216 km by 216 km) and a fine-resolution 4D-VAR inversion (36 km by 36 km) and is performed using IASI-$NH_3$ observations in April, July, and October. The hybrid approach overcomes the over-adjusting problem for high emission areas in the direct 4D-Var method and reduces the computational cost.

We use the $NH_3$ emission from 2011 NEI commonly used in regional and national simulations and assessments as the *a priori* emission. We find that the optimized $NH_3$ emission inventory differs greatly with the 2011 NEI in April. The emission in Midwest is overestimated and the emission in Southern states is underestimated in the 2011 NEI. Overall, the optimized emission is 46% higher. The optimized emission estimates in July and October are slightly higher (6.6% and 6.9%) than the 2011 NEI estimates and the spatial distribution agrees well. The IASI-$NH_3$ observations indicate a consistent underestimation of $NH_3$ emissions in California's San Joaquin Valley in all three months, however, the inverse modeling fails to properly scale up the emissions in July. The evaluation of simulation outputs against ground measurements including ambient $NH_3$ concentrations from AMoN and $NH_4^+$ wet deposition from NADP shows that the optimized $NH_3$ emission estimates improve the agreement between model outputs and independent observations, especially in April.

Application of the optimized $NH_3$ emission estimates also yields a better agreement between the simulated and observed $PM_{2.5}$ concentration in April in the Midwest region by improving the model performance on simulated $NH_4^+$ and $NO_3^-$. It is consistent with previous findings that the uncertainty in $NH_3$ emission is a key factor limiting the model performance of $PM_{2.5}$. The optimized $NH_3$ emission estimates in general increase the Nr deposition



amount and the relative importance of reduced form Nr, highlighting the importance of constraining $NH_3$ emission
estimates for accurately assessing nitrogen deposition and ecosystem health over sensitive regions.

*Data availability.* The IASI/Metop-A $NH_3$ total column Level 2 data is available at the IASI portal provided by the
AERIS data infrastructure (ULB, 2018). Independent observations for evaluation including surface $NH_3$
concentrations, $NH_4^+$ wet depositions, and speciated $PM_{2.5}$ concentrations are available from the NADP website and
Air Quality System (NADP, 2019, 2014; USEPA, 2018).

*Author contribution.* AR and YC conceived the study. YC, AR, HZ, and JK contributed to the design the method.
YC conducted the inverse modeling and data analysis. LC, PFC and MVD are responsible for the IASI NH3 data.
SC, SZ, AH, MR, MT, DH, PP, JR, AN, AP, SN, JB, KF, GC, CS, TC, AR developed the adjoint model of CMAQ.
YC prepare the manuscript, with discussions and comments from HS, AR, JK, YH, SC, SZ, JS, and GP. All authors
have given approval to the final version of the manuscript.

*Competing interests.* The authors declare that they have no conflict of interest.

*Disclaimer.* Contents of this publication are solely the responsibility of the grantee and do not necessarily represent
the official views of the supporting agencies. Further, the US government does not endorse the purchase of any
commercial products or services mentioned in the publication.

**Acknowledgments**

This publication was made possible by funding from the US EPA under grants R83588001, NASA under grant
NNX16AQ29G, and China Scholarship Council (CSC) Grant #201606010393. The authors acknowledge the AERIS
data infrastructure for providing access to the IASI data in this study. ULB has been supported by the Belgian State
Federal Office for Scientific, Technical and Cultural Affairs (Prodex arrangement IASI.FLOW). L.C. and M.V.D are
respectively research associate and postdoctoral researcher with the Belgian F.R.S-FNRS.

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

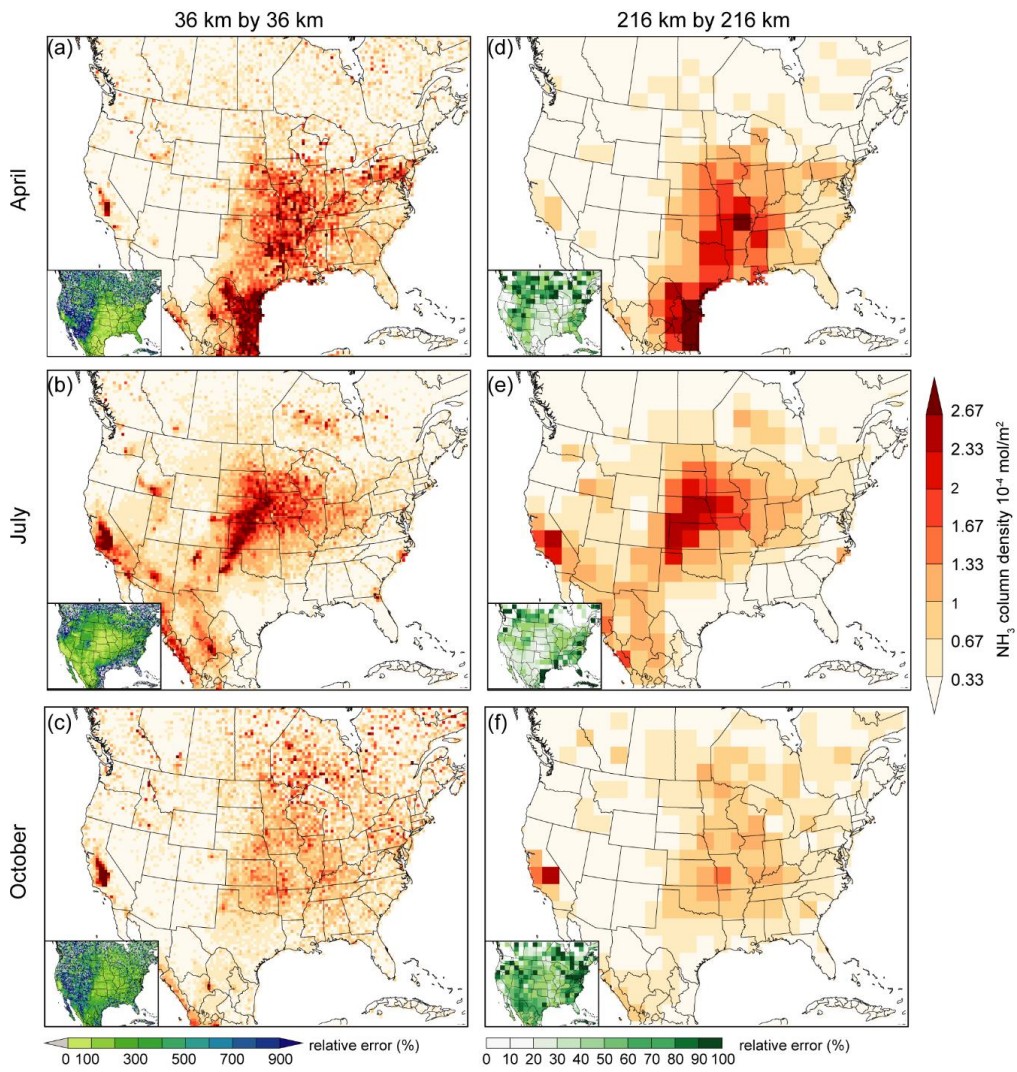

**Figure 1** IASI monthly average NH$_3$ column density in April, July, and October 2011 at 36 km by 36 km (a, b, c) and 216 km by 216 km (d, e, f) resolutions within the model simulation domain of this study. The average relative error associated with the column density is shown in the corner of each plot.

635

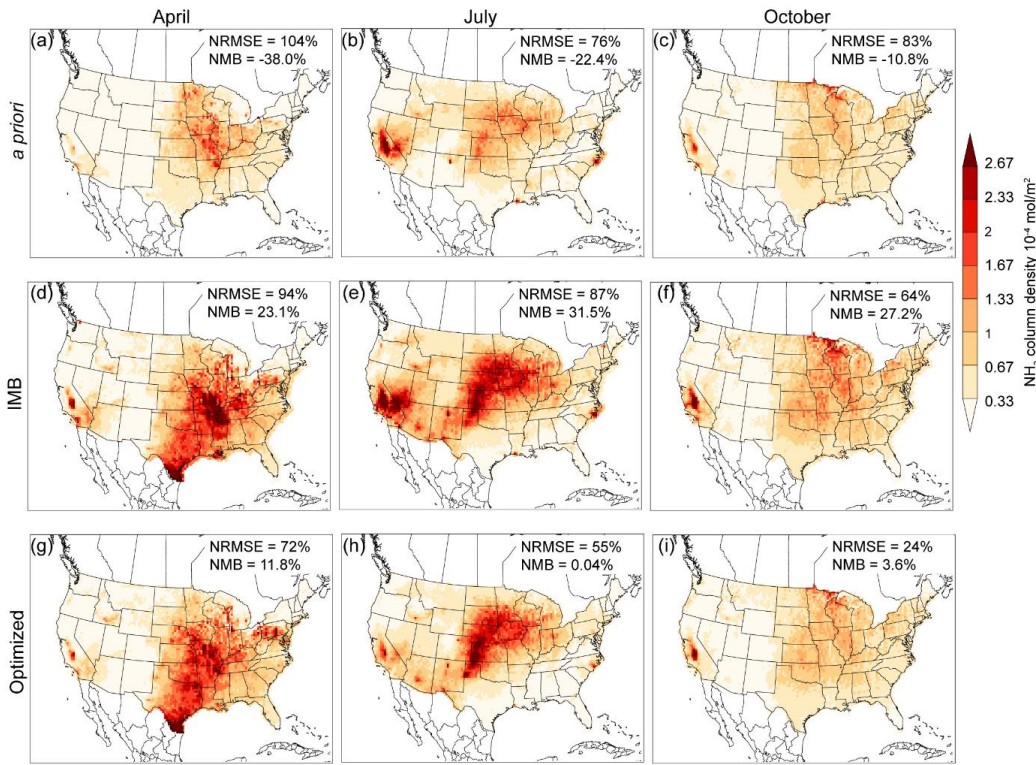

**Figure 2** CMAQ simulated monthly average NH$_3$ column density for April, July, and October 2011 using the *a priori* emissions (a, b, c), the emissions adjusted by IMB (d,e,f), and the final optimized emissions using the hybrid approach (g,h,i). For comparison with the IASI-NH$_3$ retrievals, simulated NH$_3$ columns at the passing time were derived when there are observations in that grid. Normalized root mean square error (NRMSE) and normalized mean bias (NMB) between the simulated values and IASI-NH$_3$ are provided.

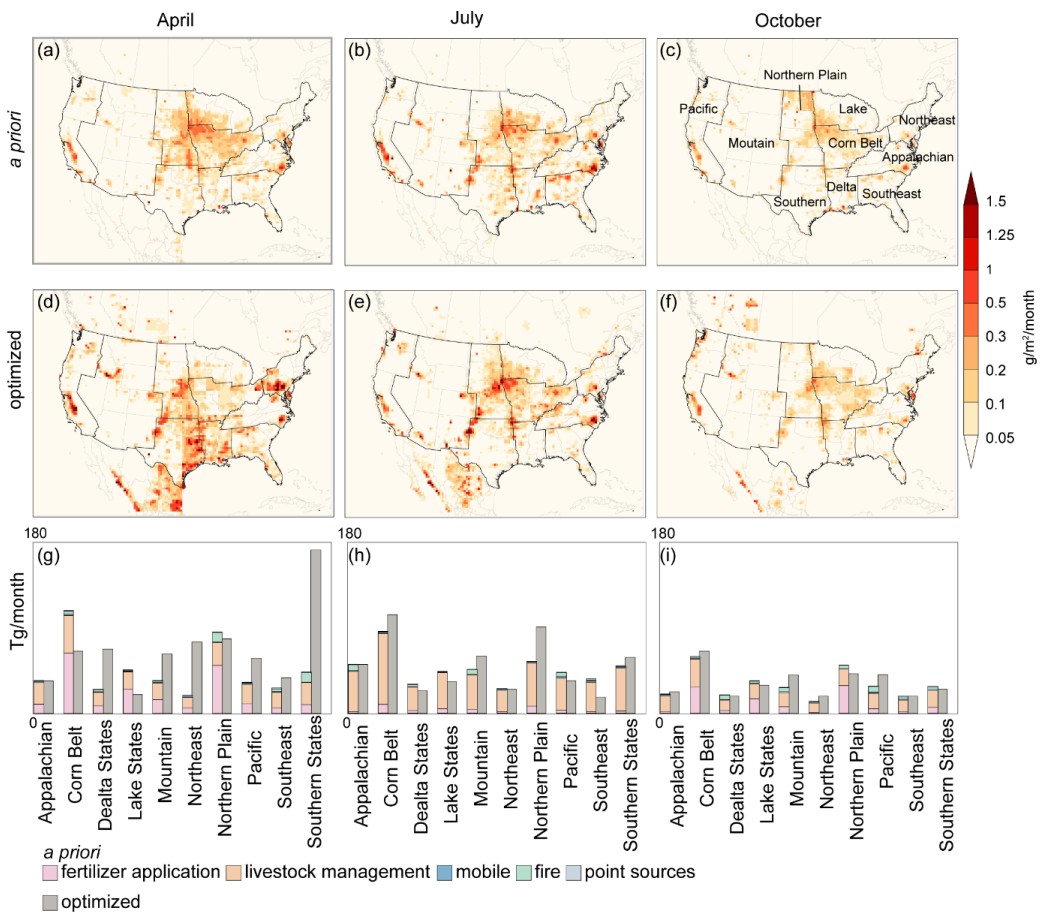

**Figure 3** The spatial distribution of monthly total NH$_3$ emission from the *a priori* (a, b, c) and optimized (d, e, f) estimates in April, July, and October. The total emission based on the *a priori* and optimized estimates are summarized for each USDA Farm Production region (g, h, i). The source contributions to total emission are shown for the *a priori* estimates.

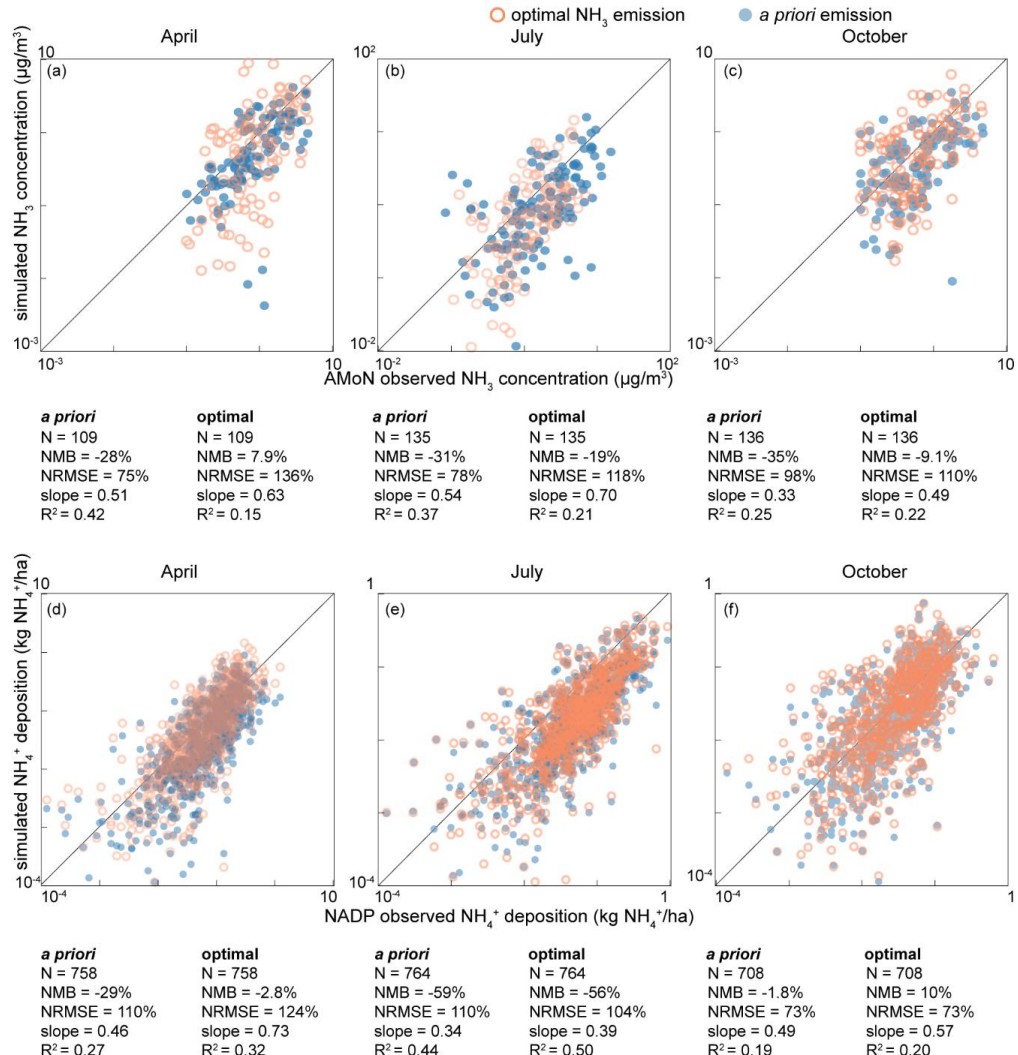

**Figure 4** Evaluation of the simulated NH₃ surface concentration (a, b, c) and NH₄⁺ wet deposition (d, e, f) against biweekly NH₃ concentration observations from AMoN and weekly NH₄⁺ wet deposition observations from NADP, respectively. The orange circles and blue dots represent comparison using the *a priori* and optimized NH₃ emission estimates, respectively. Summary statistics including sample size (N), normalized mean bias (NMB), normalized root mean square error (NRMSE), least square error regression slope and intercept, and R square ($R^2$) for all comparisons are listed below the plots.

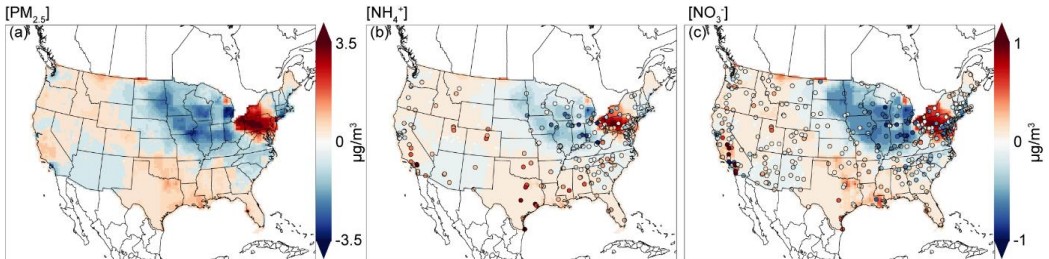

**Figure 5** The changes in monthly average PM$_{2.5}$, NH$_4^+$, and NO$_3^-$ mass concentration in April due to the NH$_3$ emission adjustment in the optimized estimates. The change is defined as conc$_{optimized}$ − conc$_{a\ priori}$, where conc$_{optimized}$ and conc$_{a\ priori}$ represents the simulated monthly average mass concentration using the optimized and *a priori* NH$_3$ emission estimates, respectively. The difference between the observed NH$_4^+$, and NO$_3^-$ mass concentration and simulated concentrations using the *a priori* NH$_3$ emission (conc$_{obs}$ − conc$_{a\ priori}$, where conc$_{obs}$ represents the observed monthly average mass concentration) are overlaid using colored dots with the same color scheme.

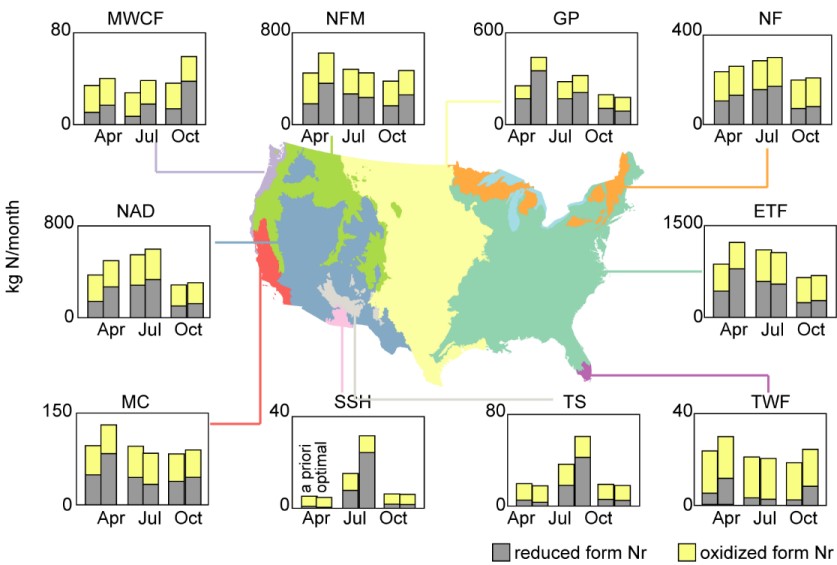

**Figure 6** The changes in the simulated monthly reactive nitrogen (Nr) deposition amount in protected areas for biodiversity conservation caused by the emission adjustment in April, July, and October. For each month, the left bar is for the a priori deposition amounts and the right bar is for the optimized deposition amounts. The deposition is grouped for 10 level I ecoregions defined by the Commission for Environmental Cooperation, including Northern Forests (NF), Great Plains (GP), Northwestern Forested Mountains (NFM), Marine West Coast Forest (MWCF), North American Deserts (NAD), Mediterranean California (MC), Southern Semiarid Highlands (SSH), Temperate Sierras (TS), and Tropical Wet Forests (TWF).

640





**Table 1** Statistical summary of the correlation between simulated monthly average $NH_4^+$ and $NO_3^-$ concentrations and observations in April[a]

| $NH_4^+$ | Midwest | | Penn | | Other | |
|---|---|---|---|---|---|---|
| | *a priori* | optimized | *a priori* | optimized | *a priori* | optimized |
| N | 47 | | 37 | | 115 | |
| NMB | 0.18 | 0.03 | 0.03 | 0.33 | -0.24 | -0.2 |
| NRMSE | 0.39 | 0.29 | 0.33 | 0.59 | 0.45 | 0.49 |
| slope | 0.52 | 0.60 | 0.47 | 0.33 | 0.74 | 0.28 |
| $R^2$ | 0.60 | 0.65 | 0.34 | 0.49 | 0.22 | 0.08 |
| $NO_3^-$ | Midwest | | Penn | | Other | |
| | *a priori* | optimized | *a priori* | optimized | *a priori* | optimized |
| N | 69 | | 38 | | 240 | |
| NMB | 0.50 | 0.22 | 0.10 | 0.58 | -0.66 | -0.69 |
| NRMSE | 0.75 | 0.51 | 0.27 | 0.72 | 0.82 | 1.03 |
| slope | 0.44 | 0.50 | 0.18 | 0.48 | 0.33 | 0.48 |
| $R^2$ | 0.76 | 0.72 | 0.31 | 0.67 | 0.13 | 0.67 |

[a] The correlation between observed concentrations and simulated ones based on *a priori* and optimized $NH_3$ emission estimates are compared. The sites are grouped as the Midwest region, Pennsylvania state and surrounding areas, and other areas.

645