# Peer review of "High-resolution Hybrid Inversion of IASI Ammonia Columns to Constrain U.S. Ammonia Emissions Using the CMAQ Adjoint Model"

_Atmospheric Chemistry and Physics, 2020_

## Referee Comment (RC1) · Anonymous Referee #1 · 23 Aug 2020

This manuscript by Chen et al. used the recently developed multiphase CMAQ adjoint model and IASI satellite total $NH_3$ column observations to constrain the monthly NEI $NH_3$ emissions at 36 km spatial resolution in April, July, and October in 2011. A hybrid, two-step optimization scheme was applied. First the NEI inventory was nudged towards the posterior values by a mass-balance approach at a much coarser grid (216 km), and then 4D-Var inversion was performed using the updated inventory as the prior. The posterior emissions were then used to drive the CMAQ model, and the simulated $NH_3$ abundance, $NH_4$ deposition, and aerosol chemical composition were evaluated against independent observational datasets. Overall this work is solid, has applied state-of-the-art satellite data and CTM tools, and could advance our limited understanding on the

emission of $NH_3$ if its methodology can be fully justified. Hopefully the paper can be further improved after addressing my comments below.

General comments

1. NEI 2011 covers the entire year continuously but this work only focused on three months, April, July, and October. Presumably the computing cost prohibited optimizing NEI for other months, but this should be discussed. Many CTM users would use multiple months or the entire year of NEI, and those three isolated months would hinder further application of the results of this work. Especially, the month of May will be a significant opportunity missed as a large fraction of fertilizer application happens in May, leading to abruptly different emission and column density dynamics relative to April and June.

2. The observation used in the inversion seems to be monthly averaged data over 36-km grid cells, and the grid average absolute error was used in the observational error covariance matrix. This may have led to the counterintuitively high values in Pennsylvania and southern Texas, as the monthly averaged grid value could have been driven by a few anomalously high observation dates, given the sparsity of IASI pixels. The error term (in Equation 1) does not include the scaling of the square root of N (the central limit theorem). As a result, if a grid cell contained only one day with extremely high values (the other days in the month were missing), it would be treated the same way as if all 30 days were those high values. Specifically, the high emissions in Pennsylvania, western New York, and east/south Texas (Fig. 3d) that were seemly driven by high IASI values in April (Fig. 1a) are hard for me to believe. It might be helpful to check IASI April data in other years, e.g., 2010 and 2012, to see if those high column abundance (and consequently high posterior emissions) are consistent.

Specific comments

Page 2, line 49: clarify which NEI it is (prior or posterior) in "NEI-based" assessments.

Page 2, lines 61-65: this sentence might fit better at the last paragraph of the introduction.

Page 5, equation 1: this is a strange statistic to calculate. As indicated a few lines above, $\Omega_0$ is the monthly arithmetic mean within a grid cell, but the $\sum(\sigma_i/\sigma_i^2)/\sum(1/\sigma_i^2)$ term is the variance-weighted mean of error. A simple standard error of the mean or standard error of the weighted mean (https://doi.org/10.1016/1352-2310(94)00210-C) might be better choices.

Page 7, lines 201-202: how justified is it to assume that the a priori covariance matrix is diagonal? The error/bias in NEI often seem spatially correlated.

Page 7, lines 203: it is important to let the readers understand if the observation vector used in the inversion is composed of single IASI pixels (level 2) or regridded maps (level 3). My impression is that the level 3 regridded IASI data were used. In that case, the single sounding detection limit of $4.8 \times 10^{15}$ is not relevant as the averaging will reduce the noise level, and it is important to consider the number of averaging per grid cell.

Page 7, line 215: the convergence criterion that $J$ decreases by less than 2% seems large and arbitrary.

Figure 2: please consider adding the residual map (IASI column-modeled column) as inserts, similar to Fig. 1.

Page 8, lines 234-235: please define the exact location of NRMSE that reduced by 98%. The high $NH_3$ observations in April in southern states seem curious and may warrant a closer investigation.

Page 9, line 277: it may be helpful to also include a priori emission totals in those three months. The posterior emission indicates that the total $NH_3$ emission decreases linearly from April to July and to October. Then what would the seasonality look like?

Page 9, line 297 and page 12, line 384: it is contradictory to claim that the hybrid inversion "overcomes the over-adjusting problem for high emission rates" and meanwhile attribute the worsening RMSE against AMoN to the emission over-adjustment problem that has supposedly been overcome. Especially the comparison between posterior and AMoN in April (Fig. 4a) seems problematic.

Table 1: the $R^2$ of 0.08 at other (also the majority of) sites between simulated $NH_4^+$ and observations in April is bothersome. The N is a reasonably large number (115), so such a low $R^2$ indicates that the model essentially lost all explanation power after the inversion. The authors are suggested to take a closer look at the April data (for other years than 2011 as well) and make sure they are representative.

Page 10, line 302-303 and page 11, lines 345-346: as CMAQ is a full 3D CTM driven by real WRF meteorology and hourly emissions, those transport should have been captured. Why not?

---

## Referee Comment (RC2) · Anonymous Referee #2 · 16 Sep 2020

This manuscript applied a hybrid inversion approach, which combines a coarseresolution mass balance inversion and a fine-resolution 4D-VAR inversion, to optimize NH3 emission estimates from the 2011 National emission inventory (2011 NEI) for the U.S. based on the satellite observations of the Infrared Atmospheric Sounding Interferometer NH3 column density (IASI-NH3) and the numerical simulations using the CMAQ v5.0 and its multiphase adjoint model. The optimized NH3 emission inventory suggests the underestimation in the 2011 NEI, especially the NH3 emission amount in April. The study demonstrated the robustness of the inversed NH3 emission inventory by evaluating the CMAQ modeling performance of ambient NH3 concentrations and NH4+ wet deposition, analyzed the potential factors accounting to the differences between the NH3 emissions in 2011 NEI and the optimized estimates, and assessed the influences of the optimized NH3 emissions to the simulations of ambient aerosol concentrations as well as to the nitrogen deposition exceedances in the U.S. The results are presented in a clear way and the manuscript stands in a good structure. I would recommend publication in Atmospheric Chemistry and Physics after consideration of the following comments.

**Specific comments**

1. The adjustment to the a priori emissions of NH3 is driven by the difference between the observed NH3 column density and the simulated one, which requires that the uncertainty in the a priori emissions is the dominant explanatory factor for the bias in the simulated NH3 column density. As we know, several factors other than NH3 emissions might affect the uncertainty of the simulated NH3 column density, such as the meteorological fields, the simulated concentrations of other related species, and even other primary emissions. The performance of the WRF model and the CMAQ model in the study are suggested to be introduced in the section 2.3. The influences of these factors on the inversion of NH3 emissions are also suggested to be discussed in the evaluation of the optimized emission estimates.

2. In section 3.3, lines 301-306: Do the outputs of the WRF/CMAQ model present the large transported plume from the central U.S. to Pennsylvania on April 14th and 15th? Do other data or analysis (such as wind observations at high altitude, trajectory analysis) support the possibility of this transport?

3. As shown in Figure 4, the optimized NH3 emission reduces the negative NMB when comparing the CMAQ outputs with AMoN NH3 concentrations, but increases the NRMSE and decreases the correlation. In my opinion, the optimized NH3 inventory does not greatly improve the agreement between CMAQ simulated NH3 concentrations and the observations. The near ground ambient NH3 concentrations might reflect more direct signal of the NH3 emissions than the NH3 column density. If the ambient NH3

measurements together with the satellite observations are used to inverse the NH3 emissions, we would obtain more reasonable optimized emission estimates.

Technical comments

1. In lines 434-436 and lines 541-542: Please add the journals which the references are submitted to.

СЗ

---

## Author Response (AR2)

We thank the reviewer for providing the thoughtful comments and suggestions. Point-to-point responses to all comments are provided below. Corresponding changes in the manuscript are described in *italics*. The inversion is re-performed in response to comments from reviewer #1. The manuscript has been updated based on the new inversion results with track changes.

**Response to Reviewer #1 comments:**

**Comment 1**

This manuscript by Chen et al. used the recently developed multiphase CMAQ adjoint model and IASI satellite total $NH_3$ column observations to constrain the monthly NEI $NH_3$ emissions at 36 km spatial resolution in April, July, and October in 2011. A hybrid, two-step optimization scheme was applied. First the NEI inventory was nudged towards the posterior values by a mass-balance approach at a much coarser grid (216 km), and then 4D-Var inversion was performed using the updated inventory as the prior. The posterior emissions were then used to drive the CMAQ model, and the simulated $NH_3$ abundance, $NH_4$ deposition, and aerosol chemical composition were evaluated against independent observational datasets. Overall this work is solid, has applied state-of-the art satellite data and CTM tools, and could advance our limited understanding on the emission of $NH_3$ if its methodology can be fully justified. Hopefully the paper can be further improved after addressing my comments below.

**Response**

We thank the reviewer for providing insightful comments. In this revision, we addressed these comments carefully. In particular, the inversion was re-performed using daily IASI-$NH_3$ averages as constrain and revised error terms. The revisions help partly resolved the over-adjustment issue we encountered in Pennsylvania and surrounding regions. Please see our point-by-point responses for details. We hope that this new version of the manuscript has addressed all the concerns raised by the reviewer.

**Comment 2**

NEI 2011 covers the entire year continuously but this work only focused on three months, April, July, and October. Presumably the computing cost prohibited optimizing NEI for other months, but this should be discussed. Many CTM users would use multiple months or the entire year of NEI, and those three isolated months would hinder further application of the results of this work. Especially, the month of May will be a significant opportunity missed as a large fraction of fertilizer application happens in May, leading to abruptly different emission and column density dynamics relative to April and June.

**Response**

We thank the reviewer for pointing this out. Yes, we focused on three months because the computational cost to run full year simulation using adjoint model is too high. The CPU time required for one-day forward and adjoint simulation is 9.5 hours and 48 hours, respectively, which means that it takes over 20,000 CPU hours to perform a full year simulation. If the inversion takes 3~5 iterations to reach the converge criteria, the CPU time can reach 60,000 to 100,000 hours. A sentence is added to line 155 to clarify that the optimization only focused on three months due to the high computational cost as follows. "*Limited by the high computational cost of adjoint-model-based inversion, the optimization is only performed for the three months selected instead of a full year.*" In addition, as explained in the sentence in line 155 to 156, the optimization was not performed for the winter months (November, December and January) because the IASI-$NH_3$ observations are too noisy to serve as a reliable constrain.

The comparison between monthly average IASI $NH_3$ column density and CMAQ simulated values using the *a priori* $NH_3$ emission inventory for all twelve months in 2011 are provided in the revised SI (**Figure S9**). Simulated $NH_3$ column densities are biased low comparing to the IASI observed ones especially from April to August. For May, the simulated $NH_3$ column densities are much lower than the IASI observations, especially in southern states (Texas and Oklahoma). Although we only performed the inverse modeling in April to represent the spring months, we expect

the emission and column density dynamics in May are similar to those in April. Sentences are added in line 287 to imply the potential low bias of NH₃ emission estimates in the NEI inventory in other months. "*Although the inversion is only applied for the three selected months, the simulated NH₃ column densities using the a priori inventory are consistently lower than the IASI-NH₃ observations in 2011 (**Figure S9**), suggesting that the NH₃ emission estimates in 2011 NEI may be biased low in other months, too.*"

Figure S9 was added to SI to provide the results of the comparison between monthly average IASI NH₃ column density and CMAQ simulated values using the *a priori* NH₃ emission inventory for all twelve months in 2011.

[Figure]

**Figure S9** Comparison between monthly average IASI NH₃ column density (a-c, g-i, m-o, s-u) and CMAQ simulated values (d-f, j-l, p-r, v-x) based on the *a priori* NH₃ emission inventory in 2011. The monthly average relative error associated with the observed IASI NH₃ column density is shown in the corner of the corresponding plots.

[Figure]

**Figure S9** (continued) Comparison between monthly average IASI NH$_3$ column density (a-c, g-i, m-o, s-u) and CMAQ simulated values (d-f, j-l, p-r, v-x) based on the *a priori* NH$_3$ emission inventory in 2011. The monthly average relative error associated with the observed IASI NH$_3$ column density is shown in the corner of the corresponding plots.
* * *
**Comment 3**

The observation used in the inversion seems to be monthly averaged data over 36-km grid cells, and the grid average absolute error was used in the observational error covariance matrix. This may have led to the counterintuitively high values in Pennsylvania and southern Texas, as the monthly averaged grid value could have been driven by a few

anomalously high observation dates, given the sparsity of IASI pixels. The error term (in Equation 1) does not include the scaling of the square root of N (the central limit theorem). As a result, if a grid cell contained only one day with extremely high values (the other days in the month were missing), it would be treated the same way as if all 30 days were those high values. Specifically, the high emissions in Pennsylvania, western New York, and east/south Texas (Fig. 3d) that were seemly driven by high IASI values in April (Fig. 1a) are hard for me to believe. It might be helpful to check IASI April data in other years, e.g., 2010 and 2012, to see if those high column abundance (and consequently high posterior emissions) are consistent.

**Response**

The reviewer's thought is well-taken. Indeed, using monthly averaged $NH_3$ column densities and averaged absolute error may lead to biased posterior emission estimates when the high averaged values are driven by high observations in several days. In response to this comment, we redid the inversion using daily observations as constraints. We also change the method to calculate the error term. A simple standard error of the mean column density in each grid was used. Please note that this was achieved by rerunning all the simulations, which was one of the main reasons we postponed the revision due date.

The specifics are described as follows.

The sentence in lines 146-147 is revised as "*The mean column density ($\Omega_o$) is calculated as the arithmetic mean of all retrievals with their centroids falling in the same grid cell, following ...*"

The sentence in line 148-153 is revised as "*The error (molec/cm$^2$) corresponding to the mean column density in each grid is calculated as:*

$$\bar{\sigma} = \sqrt{\frac{\sum(\sigma_i \times \Omega_i)^2}{n-1}}$$

*where $\bar{\sigma}$ is the mean error (molec/cm$^2$), $\Omega_i$ is the NH$_3$ column density from IASI-NH$_3$ level 2 data, $\sigma_i$ is the relative error associated with each $\Omega_i$ as reported, n is the number of retrievals within each grid cell during the defined time period. For 4D-Var inversion and IMB inversion, daily and monthly means and errors are calculated, respectively.*"

For the iterative mass balance optimization (IMB) step, the emission scaling factors are still derived at 216 km by 216 km resolution. However, in each day, only grid cells with satellite observations at 36 km by 36 km resolution are adjusted. Otherwise, the grid cells without observations at 36 km by 36 km resolution may be over-adjusted in the IMB step and there will not be enough constraint in the 4D-Var inversion to further adjust the emissions in these grid cells.

The sentences in line 178-185 are revised as "*The first step is to apply the IMB approach to adjust the a priori (2011 NEI) NH$_3$ emission at 216 km by 216 km resolution (referred to as the coarse grid cell hereafter) based on the ratio between the monthly-averaged observed and simulated NH$_3$ column density at the satellite overpassing time, iteratively. At each iteration, the emission in each 36 km by 36 km grid cell (referred to as the fine grid hereafter) is scaled by the ratio following the equation below,*

$$E_{t,i,j} = \begin{cases} \frac{\Omega_{o,m}}{\Omega_{a,m}} \times E_{a,i,j}, & IASI\ pixels\ available\ in\ grid\ cell\ i\ in\ day\ j \\ E_{a,i,j}, & no\ IASI\ pixels\ in\ grid\ cell\ i\ in\ day\ j \end{cases} \qquad (2)$$

*where $E_{t,i,j}$ and $E_{a,i,j}$ are the new and a priori emission estimates in fine grid cell i within the coarse grid cell on the jth day of the month, respectively. $\Omega_{o,m}$ and $\Omega_{a,m}$ are the monthly-averaged observed and simulated NH$_3$ column density in coarse grid cells, respectively. It is a modified version of IMB optimization performed in previous studies (Li et al., 2019; Cooper et al., 2017; Martin et al., 2003). The emissions in grid cells without IASI retrievals are kept unchanged to avoid over-adjustment.*"

The sentence in line 190 is revised as "*The final scaling factor ($\varepsilon_0$) for each grid cell is the multiplication of the scaling factors derived at each iteration.*"

For the 4D-Var inversion, daily mean column density from the IASI-NH$_3$ observations are used as constraints. Daily emission scaling factors are derived through optimization.

The sentences in lines 119-210 are revised as below.

"

$$J = \gamma(\varepsilon - \varepsilon_0)^T S_a^{-1}(\varepsilon - \varepsilon_0) + \left(\Omega_{o,d} - F(\varepsilon)\right)^T S_o^{-1}(\Omega_{o,d} - F(\varepsilon)) \tag{3}$$

*$\varepsilon$ is the daily emission scaling factor to be optimized at each iteration where $\varepsilon = log\left(E_t/E_a\right)$ on the 36 km by 36 km CMAQ grid, consisting of 6104 elements overland grid cells in CONUS. $\Omega_{o,d}$ is daily-averaged IASI-NH$_3$ column densities and F(ε) is CMAQ simulated NH$_3$ column density sampled at the satellite passing time if there is at least one IASI-NH$_3$ retrieval in that grid cell. $S_a$ and $S_o$ are error covariance matrices for the a priori emission estimates and IASI-NH$_3$ retrievals, respectively. The two matrices are assumed to be diagonal. For $S_o$, the simple standard error corresponding to $\Omega_{o,d}$ is used to represent the observational error (Equation (1)). Our test shows that negative $\Omega_{o,d}$ will lead to a continuous decrease in the adjusted emission for the grid cell because modeled column density cannot become negative. To limit the influence of these negative $\Omega_{o,d}$, their original weights are multiplied by 0.01. For $S_a$, the uncertainty in each grid cell is assumed to be 100% of the a priori emissions. γ is the regularization factor balancing the relative contribution of the a priori emission inventory and IASI-NH$_3$ retrievals to the J value. γ is chosen to be 800 for April and 500 for July and October based on the L-curve criteria (Hansen, 1999) (**Figure S5**).*"

Using daily mean IASI-NH$_3$ column densities as constraints do help alleviate the over-adjustment in Pennsylvania in April. The *posterior* emission estimate in Pennsylvania is 127% higher than the *a priori* estimates using daily means as constraint, whereas 717% higher when using monthly means. For Texas, the difference is smaller (237% higher using daily means versus 335% higher using monthly means) because high IASI-NH$_3$ column densities were observed on many days, possibly because of the warmer weather condition and earlier fertilizer application activities in 2011. Please refer to the response to **Comment 11** for a detailed discussion. Again, we thank the reviewer for providing this insightful comment on the inversion method.
* * *
**Comment 4**

Page 2, line 49: clarify which NEI it is (prior or posterior) in "NEI-based" assessments.

**Response**

Thanks for the suggestion. The sentence is revised as "*The model results suggest that the estimated contribution of ammonium nitrate would be biased high in a priori NEI-based assessments.*"
* * *
**Comment 5**

Page 2, lines 61-65: this sentence might fit better at the last paragraph of the introduction.

**Response**

We thank the reviewer for this suggestion. The sentence in lines 61-65 is moved to the beginning of the last paragraph of the introduction. The last paragraph is revised as "*This work utilizes satellite observations from the IASI NH$_3$ column density measurements (IASI-NH$_3$) (Clarisse et al., 2009;Van Damme et al., 2017), to provide a high-resolution, optimized NH$_3$ emission inventory for the U.S. developed using an adjoint inverse modeling technique (Li et al., 2019), the robustness of which is demonstrated by evaluation against multiple independent in-situ measurements. The IASI-NH$_3$ dataset was applied to optimize NH$_3$ emission estimates from the 2011 National Emission Inventory (NEI 2011) using CMAQ and its adjoint model at a 36 km×36 km resolution. ...*"

IASI is spelled out at its first appearance in line 70 as "*Several studies have utilized NH$_3$ column density retrieved from the Infrared Atmospheric Sounding Interferometer (IASI) (Clarisse et al., 2009; Van Damme et al., 2015b) …*"

**Comment 6**

Page 5, equation 1: this is a strange statistic to calculate. As indicated a few lines above, $\Omega_0$ is the monthly arithmetic mean within a grid cell, but the $\sum(\sigma_i/\sigma_i^2)/\sum(1/\sigma_i^2)$ term is the variance-weighted mean of error. A simple standard error of the mean or standard error of the weighted mean (https://doi.org/10.1016/1352-2310(94)00210-C) might be better choices.

**Response**

In response to this comment, the error term is changed to a simple standard error of the daily mean in the revised manuscript, and the simulations are re-performed with the revised error covariance matrices. The results are updated throughout the text.

The sentence in line 148-153 is revised as "*The error (molec/cm$^2$) corresponding to mean column density in each grid cell is calculated as:*

$$\bar{\sigma} = \sqrt{\frac{\sum(\sigma_i \times \Omega_i)^2}{n-1}}$$

*where $\bar{\sigma}$ is the mean error (molec/cm$^2$), $\Omega_i$ is the NH$_3$ column density from IASI-NH$_3$ level 2 data, $\sigma_i$ is the relative error associated with each $\Omega_i$ as reported, n is the number of retrievals within each grid cell during the defined time period. For 4D-Var inversion and IMB inversion, daily and monthly means and errors are calculated, respectively.*"

**Comment 7**

Page 7, lines 201-202: how justified is it to assume that the a priori covariance matrix is diagonal? The error/bias in NEI often seem spatially correlated.

**Response**

Thank you for raising this concern. The error covariance matrix is assumed to be diagonal because there is no data available to estimate the spatial correlation of errors in NH$_3$ emission estimates. Including non-diagonal terms to the *a priori* covariance matrix, therefore, may further introduce uncertainties in the inverse modeling. The sentence in line 201-202 is revised to clarify the reason why the *a priori* covariance matrix is assumed to be diagonal as follow. "*With limited information on the spatial correlation of the error covariance, the two matrices are assumed to be diagonal (Paulot et al., 2014; Zhu et al., 2013).*"

**References**

Paulot, F., Jacob, D.J., Pinder, R.W., Bash, J.O., Travis, K., Henze, D.K.: Ammonia emissions in the United States, European Union, and China derived by high-resolution inversion of ammonium wet deposition data: Interpretation with a new agricultural emissions inventory (MASAGE_NH$_3$). J. Geophys. Res. Atmos. 119, 4343-4364, https://doi.org/10.1002/2013JD021130, 2014.

Zhu, L., Henze, D. K., Cady-Pereira, K. E., Shephard, M. W., Luo, M., Pinder, R. W., Bash, J. O., and Jeong, G. R.: Constraining U.S. ammonia emissions using TES remote sensing observations and the GEOS-Chem adjoint model, J. Geophys. Res. Atmos., 118, 3355-3368, https://doi.org/10.1002/jgrd.50166, 2013.

**Comment 8**

Page 7, lines 203: it is important to let the readers understand if the observation vector used in the inversion is composed of single IASI pixels (level 2) or regridded maps (level 3). My impression is that the level 3 regridded IASI

data were used. In that case, the single sounding detection limit of $4.8\times10^{15}$ is not relevant as the averaging will reduce the noise level, and it is important to consider the number of averaging per grid cell.

**Response**

Yes, level 3 regrided IASI data is used in the inversion. In response to comment 3, the inversion was re-performed and daily means regrided at 36 km 36 km resolution were used as constraints in the 4D-Var inversion. The reviewer is right that the single sounding detection limit is higher than the actual noise level when pixels are averaged. We no longer add a detection limit to the error covariance $S_o$. The simulations are re-performed with the revised error covariance matrices, and the results are updated throughout the text.

This sentence in lines 203-205, "*To reduce the influence of retrievals close to or below the detection limit, an estimated detection limit of $4.8\times10^{15}$ molecules/cm$^2$ is added to the $S_o$ (Dammers et al., 2019)*", is deleted.

**Comment 9**

Page 7, line 215: the convergence criterion that J decreases by less than 2% seems large and arbitrary.

**Response**

The convergence criterion was chosen following previous inverse modeling studies to optimize NH$_3$ emission estimates. Citations are added in the sentence in line 215 to clarify the choice of the convergence criterion. "*The iteration process is terminated when the decrease in J is less than 2% or the local minimum is reached (Li et al., 2019; Zhu et al., 2013).*"

**References**

Li, C., Martin, R. V., Shephard, M. W., Cady-Pereira, K., Cooper, M. J., Kaiser, J., Lee, C. J., Zhang, L., and Henze, D. K.: Assessing the Iterative Finite Difference Mass Balance and 4D-Var Methods to drive ammonia emissions over North America using synthetic observations, J. Geophys. Res. Atmos., 124, 4222-4236, https://doi.org/10.1029/2018jd030183, 2019.

Zhu, L., Henze, D. K., Cady-Pereira, K. E., Shephard, M. W., Luo, M., Pinder, R. W., Bash, J. O., and Jeong, G. R.: Constraining U.S. ammonia emissions using TES remote sensing observations and the GEOS-Chem adjoint model, J. Geophys. Res. Atmos., 118, 3355-3368, https://doi.org/10.1002/jgrd.50166, 2013.

**Comment 10**

Figure 2: please consider adding the residual map (IASI column-modeled column) as inserts, similar to Fig. 1.

**Response**

Residual maps are inserted as suggested by the reviewer. Figure 2 is revised as follows,

[Figure]

**Figure 2** CMAQ simulated monthly average NH₃ column density for April, July, and October 2011 using the *a priori* emissions (a, b, c), the emissions adjusted by IMB (d,e,f), and the final optimized emissions using the hybrid approach (g,h,i). For comparison with the IASI-NH₃ retrievals, simulated NH₃ columns at the passing time are derived when there are observations in that grid cell. Normalized root mean square error (NRMSE) and normalized mean bias (NMB) between the simulated values and IASI-NH₃ are provided. Residue map (IASI-NH₃ – simulated NH₃ column densities) is shown in the corner of each plot.
* * *
**Comment 11**

Page 8, lines 234-235: please define the exact location of NRMSE that reduced by 98%. The high NH₃ observations in April in southern states seem curious and may warrant a closer investigation.

**Response**

By "southern states" we are referring to the states in the southern region defined by the USDA Farm Production region, which includes Texas and Oklahoma. In the revised simulation, the NRMSE in the southern states was reduced by 50% instead of 98% with the optimized NH₃ emission estimates. The sentence in lines 234-235 is revised as follows "*The optimized NH₃ emission successfully captures the high NH₃ column density in southern states (Texas and Oklahoma), reducing the NRMSE by half in that region.*"

The enhanced NH$_3$ emissions in the southern states in the optimized emission estimates are more consistent with the total NH$_3$ emission estimates when air-surface bidirectional exchange is considered (Shen et al., 2020). The ratio between NH$_3$ emission estimates in southern states and total NH$_3$ emission within CONUS is 20% and 18% in the optimized estimates and estimates including NH$_3$ bidirectional exchange in 2011, respectively. In comparison, the ratio is only 10% in the *a priori* NEI estimates.

The comparison of IASI-NH$_3$ in 2011 and adjacent years shows a substantial variation in the retrieved NH$_3$ column densities in the southern states. The NH$_3$ observations are the highest in 2011 and the lowest in 2010 in April and May. The difference coincides with the higher surface temperature in the southern states in these two months. NH$_3$ volatilization from agricultural lands will increase under warmer conditions (Shen et al., 2020).

The pieces of evidence mentioned above are incorporated in the discussion to support the increased NH$_3$ emission in southern states in the optimized estimates as follows. The sentences in line 261-263 are revised as "*The higher NH$_3$ emission estimates in the southern states are driven by the enhanced NH$_3$ column densities from IASI over that region. IASI-NH$_3$ column densities are higher in 2011 than those in adjacent years (**Figure S7**), which coincides with the higher surface temperature observed in 2011 (NOAA 2019)(**Figure S8**). NH$_3$ emission will increase due to enhanced NH$_3$ volatilization from agricultural lands under warmer conditions (Bash et al., 2013; Shen et al., 2020). In fact, the optimized NH$_3$ emission pattern in April is more consistent with the spatial pattern of inorganic nitrogen fertilizer estimated based on plant demand (Cooter et al., 2012). NH$_3$ emission in 2011 estimated by CMAQ with NH$_3$ bidirectional exchange model also predicted higher NH$_3$ emission in the southern states (Shen et al., 2020). The ratio between NH$_3$ emission estimates in southern states and that within CONUS is 26% and 18% in the optimized estimates and estimates including NH$_3$ bidirectional exchange, respectively. In comparison, the ratio is only 10% in the a priori NEI estimates, suggesting a potential low bias in 2011 NEI.*"

Two figures were added to SI as follows to provide the IASI-NH$_3$ column densities for 2010, 2011, and 2012 and surface temperature maps for these three years.

[Figure]

**Figure S7** Monthly averaged IASI-NH3 column densities in April and May from 2010 to 2012. The satellite retrievals are regridded at 36 km by 36 km resolution.

[Figure]

**Figure S8** The monthly averaged surface temperature in April and May from 2010 to 2012.


*plume at a higher altitude from the central U.S. to Pennsylvania (**Figure S10 and Figure S11**), which is not measured by ground observations at AMoN sites at biweekly resolution. If that is the case, the hybrid inverse modeling framework would have difficulties in reproducing the long-range transport contribution for two reasons. First, local emissions in Pennsylvania would be enhanced in the IMB inversion and inter-grid transport were neglected at 216 km by 216 km resolution. Second, the following 4D-Var inversion very likely reached a local optimal by adjusting emissions from local and surrounding grid cells near the observed hotspot rather than grid cells at distance. Besides, the IASI-NH₃ column densities may be overestimated because vertical profiles with highest concentrations near the surface were assumed in the retrieval process (Whitburn et al., 2016).*"

The limitation is addressed by adding the following sentence in line 315: "*... in most of the CONUS, except in Pennsylvania and surrounding regions in April. The hybrid inverse modeling technique may over-adjust local emissions in hotspots dominated by long-range transport.*"

The sentence in line 385 is also revised as follow: "*The hybrid approach overcomes the over-adjusting problem for high emission areas in the direct 4D-Var method and reduces the computational cost, but it may introduce over-adjustment in special cases where the $NH_3$ abundance is dominated by transport instead of local emissions.*"

**Comment 14**

Table 1: the $R^2$ of 0.08 at other (also the majority of) sites between simulated $NH_4^+$ and observations in April is bothersome. The N is a reasonably large number (115), so such a low $R^2$ indicates that the model essentially lost all explanation power after the inversion. The authors are suggested to take a closer look at the April data (for other years than 2011 as well) and make sure they are representative.

**Response**

When checking the data of Table 1 multiple calculation errors are found. We sincerely apologize for the mistakes. The $R^2$ at other sites between simulated $NH_4^+$ and observations in April is 0.26 instead of 0.08.

Both corrected Table 1 for the initial submission version and revised Table 1 based on new optimizing scaling factors are provided below for comparison. The optimized $NH_3$ emission estimates still exacerbate the high bias in the Pennsylvania state and surrounding areas, but the magnitude is significantly reduced comparing to the initial version. The high IASI-NH₃ observations in April in Pennsylvania state was driven by high retrievals in a few days and using daily means instead of monthly means as constraints helped avoid the artificial high bias in the days without observations. We thank the reviewer again for the critical suggestion.

By comparing the satellite data in different years, we find that IASI-NH₃ column densities in April are higher in 2011 than in 2010 and 2012, however, it is common to have high variations in the column densities in adjacent years and months (Figure S7). We believe the IASI-NH₃ observations in 2011 show a reasonable pattern of NH₃ column densities considering the variations in meteorological conditions and emission activities. The over-adjustment in Pennsylvania and the surrounding region is possibly due to the tendency of this hybrid inverse modeling technique to over-adjust local emissions when long-range transport contributed to the high abundance of NH₃ in that region. Please refer to the response to **Comment 13** for a detailed explanation.

**Table 1 (corrected version for initial submission)** Statistical summary of the correlation between simulated monthly average $NH_4^+$ and $NO_3^-$ concentrations and observations in April[a]

| $NH_4^+$ | Midwest | | Penn | | Other | |
|---|---|---|---|---|---|---|
| | *a priori* | optimized | *a priori* | optimized | *a priori* | optimized |
| N | 47 | | 37 | | 115 | |
| NMB | 0.27 | 0.07 | 0.00 | 0.48 | -0.35 | -0.36 |
| NRMSE | 0.40 | 0.14 | 0.28 | 0.42 | 0.45 | 0.32 |

| | Midwest | | Penn | | Other | |
|---|---|---|---|---|---|---|
| | *a priori* | optimized | *a priori* | optimized | *a priori* | optimized |
| slope | 0.52 | 0.58 | 0.41 | 0.32 | 0.60 | 0.56 |
| $R^2$ | 0.57 | 0.62 | 0.24 | 0.36 | 0.25 | 0.26 |
| $NO_3^-$ | Midwest | | Penn | | Other | |
| | *a priori* | optimized | *a priori* | optimized | *a priori* | optimized |
| N | 69 | | 38 | | 240 | |
| NMB | 0.64 | 0.29 | 0.25 | 1.40 | -0.39 | -0.41 |
| NRMSE | 0.96 | 0.66 | 0.66 | 1.73 | 0.63 | 0.80 |
| slope | 0.44 | 0.50 | 0.29 | 0.48 | 0.62 | 0.33 |
| $R^2$ | 0.76 | 0.73 | 0.33 | 0.67 | 0.28 | 0.13 |

[a] The correlation between observed concentrations and simulated ones based on *a priori* and optimized $NH_3$ emission estimates are compared. The sites are grouped as the Midwest region, Pennsylvania state and surrounding areas, and other areas.

**Table 1 (revised version)** Statistical summary of the correlation between simulated monthly average $NH_4^+$ and $NO_3^-$ concentrations and observations in April[a]

| $NH_4^+$ | Midwest | | Penn | | Other | |
|---|---|---|---|---|---|---|
| | *a priori* | optimized | *a priori* | optimized | *a priori* | optimized |
| N | 47 | | 37 | | 115 | |
| NMB | 0.27 | 0.22 | 0.00 | 0.07 | -0.35 | -0.35 |
| NRMSE | 0.40 | 0.35 | 0.28 | 0.30 | 0.45 | 0.44 |
| slope | 0.52 | 0.54 | 0.41 | 0.39 | 0.60 | 0.65 |
| $R^2$ | 0.57 | 0.65 | 0.24 | 0.18 | 0.25 | 0.28 |
| $NO_3^-$ | Midwest | | Penn | | Other | |
| | *a priori* | optimized | *a priori* | optimized | *a priori* | optimized |
| N | 69 | | 38 | | 240 | |
| NMB | 0.64 | 0.55 | 0.25 | 0.43 | -0.39 | -0.38 |
| NRMSE | 0.96 | 0.88 | 0.66 | 0.73 | 0.63 | 0.65 |
| slope | 0.44 | 0.46 | 0.29 | 0.29 | 0.62 | 0.55 |
| $R^2$ | 0.76 | 0.78 | 0.33 | 0.31 | 0.28 | 0.25 |

[a] The correlation between observed concentrations and simulated ones based on *a priori* and optimized $NH_3$ emission estimates are compared. The sites are grouped as the Midwest region, Pennsylvania state and surrounding areas, and other areas.

**Comment 15**

Page 10, line 302-303 and page 11, lines 345-346: as CMAQ is a full 3D CTM driven by real WRF meteorology and hourly emissions, those transport should have been captured. Why not?

**Response**

The hybrid inverse modeling technique is a statistical optimization technique that takes into account the chemistry and physics of the CTM. The system is underdetermined because the model freedom far exceeds the number of satellite observations available. The forward CMAQ model can indeed capture long-range transport with real WRF meteorology and hourly emissions. However, instead of solving for the global optimal, the inversion can adjust emissions from local and surrounding grids instead of grids at distance to achieve a local minimum of the cost function. Besides, in our case of over-adjustment in Pennsylvania, local emissions were enhanced in the IMB inversion and inter-grid transport was neglected at 216 km by 216 km resolution.

It is a limitation of this hybrid inverse modeling method that local emissions may be over-adjusted when the satellite observed hotspots were dominated by long-range transport. The limitation is clarified and addressed in the revised manuscript as follows.

The sentences in lines 300 – 308 are revised as "*There is a greater improvement at the high concentration end than the low concentration end because both IASI satellite and the passive samplers at the AMoN sites have higher uncertainties in areas with low $NH_3$ abundance (Van Damme et al., 2015a; Puchalski et al., 2011). Yet, the NRMSE gets higher and $R^2$ gets lower in April, indicating a higher spatial variation in the residuals. There is an over-adjustment for sites in Pennsylvania in April where there is a hotspot observed by IASI in April 14th and 15th. The hotspot possibly came from a large transported plume at a higher altitude from the central U.S. to Pennsylvania (**Figure S10 and Figure S11**), which is not measured by ground observations at AMoN sites at biweekly resolution. If that is the case, the hybrid inverse modeling framework would have difficulties in reproducing the long-range transport contribution for two reasons. First, local emissions in Pennsylvania would be enhanced in the IMB inversion and inter-grid transport were neglected at 216 km by 216 km resolution. Second, the following 4D-Var inversion very likely reached a local optimal by adjusting emissions from local and surrounding grid cells near the observed hotspot rather than grid cells at distance. Besides, the IASI-$NH_3$ column densities may be overestimated because vertical profiles with highest concentrations near the surface were assumed in the retrieval process (Whitburn et al., 2016).*"

A sentence is added in line 315: *"… in most of the CONUS, except in Pennsylvania and surrounding regions in April. The hybrid inverse modeling technique possibly over-adjusts local emissions in hotspots dominated by long-range transport.*"

**Response to Reviewer #2 comments:**

**Comment 1**

This manuscript applied a hybrid inversion approach, which combines a coarse resolution mass balance inversion and a fine-resolution 4D-VAR inversion, to optimize $NH_3$ emission estimates from the 2011 National emission inventory (2011 NEI) for the U.S. based on the satellite observations of the Infrared Atmospheric Sounding Interferometer $NH_3$ column density (IASI-$NH_3$) and the numerical simulations using the CMAQ v5.0 and its multiphase adjoint model. The optimized $NH_3$ emission inventory suggests the underestimation in the 2011 NEI, especially the $NH_3$ emission amount in April. The study demonstrated the robustness of the inversed $NH_3$ emission inventory by evaluating the CMAQ modeling performance of ambient $NH_3$ concentrations and $NH_4^+$ wet deposition, analyzed the potential factors accounting to the differences between the $NH_3$ emissions in 2011 NEI and the optimized estimates, and assessed the influences of the optimized $NH_3$ emissions to the simulations of ambient aerosol concentrations as well as to the nitrogen deposition exceedances in the U.S. The results are presented in a clear way and the manuscript stands in a good structure. I would recommend publication in Atmospheric Chemistry and Physics after consideration of the following comments.

**Response**

We thank the reviewer for the comments and valuable suggestions. The detailed responses can be seen below.

**Comment 2**

Specific comments

1. The adjustment to the a priori emissions of $NH_3$ is driven by the difference between the observed $NH_3$ column density and the simulated one, which requires that the uncertainty in the a priori emissions is the dominant explanatory factor for the bias in the simulated $NH_3$ column density. As we know, several factors other than $NH_3$ emissions might affect the uncertainty of the simulated $NH_3$ column density, such as the meteorological fields, the simulated concentrations of other related species, and even other primary emissions. The performance of the WRF model and the CMAQ model in the study are suggested to be introduced in the section 2.3. The influences of these factors on the inversion of $NH_3$ emissions are also suggested to be discussed in the evaluation of the optimized emission estimates.

**Response**

We agree with the reviewer that the performance of the inversion will also be influenced by uncertainties and biases in the WRF and the CMAQ model. The model performance of the two models are added in the manuscript as suggested by the reviewer.

The WRF model performance is evaluated by comparing simulated wind speed, temperature, and humidity against surface observations. In general, the WRF simulated meteorological fields agree well with the observations, except for a slight overestimation of wind speed. The CMAQ model performance for simulating gas-particle partitioning of semi-volatile species and reactive nitrogen deposition has been evaluated in detailed in our previous papers using the same input data and model configuration (Chen et al., 2019; Chen et al., 2020). There is a consistent low bias in simulated $NH_3$ and $NH_4^+$ concentrations indicating that the $NH_3$ emission estimates are biased low. Most of the observation-simulation data pairs for $\varepsilon(NH_4^+)$ scatter within the 0.5 to 2 range, and there is no significant systematic bias found in $\varepsilon(NH_4^+)$. Larger biases were found for locations with low relative humidity, low $NH_3$ and $NO_x$ emissions, or significant dust emissions (Chen et al., 2019). For deposition evaluation, both dry and wet deposition amount are biased low, further indicating a possible low bias in $NH_3$ emission estimates. Besides, the biases in gas-particle partitioning ratio and precipitation amounts also affect the model performance (Chen et al., 2020). The most relevant evaluations including the gas-particle partitioning of $NH_3$ and $NH_4^+$ ($\varepsilon(NH_4^+)$, defined as the molar ratio of $NH_4^+$ to the sum of $NH_3$ and $NH_4^+$), as well as deposition of $NH_4^+$ are provided in the supporting information.

A sentence describing the WRF model performance is added in line 170 as follows. "*The simulated meteorological fields show good agreement with surface observations (Figure S2) (NOAA, 2020).*"

Sentences describing CMAQ model evaluation results are added in line 174, section 2.3, as follows. "*To evaluation CMAQ model performance, the simulated gas-particle partitioning ratio of NH$_3$-NH$_4^+$ and NH$_4^+$ deposition is compared with observations from AMoN, Clean Air Status and Trends Network (CASTNET), and National Atmospheric Deposition Program (NADP) (**Figure S3 and Figure S4**). CMAQ captures the overall spatial pattern of these governing processes for atmospheric NH$_3$ abundance, considering the uncertainties in emissions, model parameters, and meteorological fields. Expanded evaluation of CMAQ model performance in simulating gas-particle partitioning and nitrogen deposition has been conducted in previous studies (Chen et al, 2019; Chen et al., 2020).*"

Sentences are added in the discussion to address the impacts of uncertainties from the WRF and the CMAQ model as follows.

Sentences are added in line 284 as follows. "*Besides the a priori emission inventory and observational constraints, the inversion performance will also be affected by other processes (e. g., gas-particle partition, transport, cloud and precipitation, and dry and wet deposition) governing the atmospheric abundance of NH$_3$. Future works refining the pertinent processes will also help improve the optimized NH$_3$ emission estimates.*"

A sentence is added in line 313 as follows. "*A better representation of the cloud, precipitation, and deposition processes in the WRF and the CMAQ model is needed to close the gap between simulated and observed NH$_4^+$ deposition amount.*"

Figures showing the WRF and CMAQ performance were added to SI as follows,

[Figure]

**Figure S2** Model evaluation for WRF simulated meteorological fields against TDL hourly observations for April, July, and October. The bias and RMSE are listed below each plot.

[Figure]

**Figure S3** Model evaluation for CMAQ simulated bi-weekly average surface concentrations of $NH_3$ (a), $NH_4^+$ (b), and the gas-particle partitioning ratios, $\varepsilon(NH_4^+)$ (c) against observations from collocated AMoN (Ammonia Monitoring Network) and CASTNET (Clean Air Status and Trends Network) sites. Overlay of annual mean $\varepsilon(NH_4^+)$ based on simulated (color map) and observed (colored dots) concentrations are also plotted (d). The 1:1 line (solid black line), data range line (dashed back line with ratio labeled) and regression line (red) is also plotted. Number of data points (N), NMB, and NRMSE are provided along each plot.

[Figure]

**Figure S4** Model evaluation for CMAQ simulated wet (a and b) and dry (c) deposition of $NH_4^+$ against observations from the NADP (National Atmospheric Deposition Program) and the CASTNET (Clean Air Status and Trends) Network. Overlay of annual $NH_4^+$ wet deposition based on simulated (color map) and observed (colored dots) amount are also plotted (d). The scatter plots show the comparison between CMAQ predicted and observed annual dry, wet, and total deposition amounts, with the blue line showing the linear regression line. The 1:1 line (solid black line), data range line (dashed back line with ratio labeled) and regression line (red) is also plotted. Number of data points (N), NMB, and NRMSE are provided along each plot. For wet deposition, the CMAQ model performance with (a) and without (b) precipitation adjustment are evaluated.

**Response**

The spatial pattern of CMAQ simulated $NH_3$ column density does not present similar patterns observed by the IASI satellite on April 14[th] and 15[th], even using optimized $NH_3$ emissions as input. This is probably because the optimized results failed to capture long-range transport contribution and over-adjusted local emissions in Pennsylvania.

Although the Atmospheric Infrared Sounder (AIRS) and the Tropospheric Emission Spectrometer (TES) also measures $NH_3$ column densities in 2011, it is hard to derive daily spatial pattern in the CONUS. For AIRS, only monthly level 3 data has been developed at this moment and the coverage is poor in northeastern U.S. For TES, the satellite swath is too narrow to provide complete daily coverage for CONUS.

In the revision, we performed a trajectory analysis using NOAA HYSPLIT model driven by meteorological fields forecasted by the North American Mesoscale Forecast System (NAM) at 12 km by 12 km resolution. Forward trajectory simulation was performed for April 13[th] to 15[th] with a source located in Oklahoma at surface level (37.0 N, 94.7 W). Backward trajectory simulation was performed for April 15[th] with a receptor located in Pennsylvania (40.9 N, 77.6 W) at both surface level and elevated level (5 km). The forward air parcel trajectories show that long-range transport toward northern and northeastern regions occurred on April 14[th] and 15[th]. The backward air parcel trajectories also show that $NH_3$ in elevated height may came in from the central U.S.

[Figure]

**Figure S11** Forward and backward trajectory analysis generated from the NOAA HYSPLIT model. The location of the source (forward) and receptor (backward) are shown as stars in the figures. The starting time of each trajectory is 1 hour apart, from 00:00 to 24:00 local time on each day.

Again, the long-range transport contribution is our speculation based on the IASI-$NH_3$ spatial distribution. Although the trajectory analysis partially supports our speculation, the high IASI-$NH_3$ column densities on April 14[th] and 15[th]

warrants further investigation. In the revised manuscript, we further clarified that the long-range transport is our hypothesis to explain the discrepancy between IASI-NH$_3$ and surface observations in Pennsylvania for April 2011.

The sentences in lines 300 – 308 is revised as follows. "*There is an over-adjustment for sites in Pennsylvania in April where there is a hotspot observed by IASI in April 14$^{th}$ and 15$^{th}$. The hotspot possibly came from a large transported plume at a higher altitude from the central U.S. to Pennsylvania (**Figure S10 and Figure S11**), which is not measured by ground observations at AMoN sites at biweekly resolution. If that is the case, the hybrid inverse modeling framework would have difficulties in reproducing the long-range transport contribution for two reasons. First, local emissions in Pennsylvania would be enhanced in the IMB inversion and inter-grid transport were neglected at 216 km by 216 km resolution. Second, the following 4D-Var inversion very likely reached a local optimal by adjusting emissions from local and surrounding grid cells near the observed hotspot rather than grid cells at distance. Besides, the IASI-NH$_3$ column densities may be overestimated because vertical profiles with highest concentrations near the surface were assumed in the retrieval process (Whitburn et al., 2016).*"

Figure S11 showing the trajectory analysis results is added to the SI.
* * *
**Comment 4**

3. As shown in Figure 4, the optimized NH$_3$ emission reduces the negative NMB when comparing the CMAQ outputs with AMoN NH$_3$ concentrations, but increases the NRMSE and decreases the correlation. In my opinion, the optimized NH$_3$ inventory does not greatly improve the agreement between CMAQ simulated NH$_3$ concentrations and the observations. The near ground ambient NH$_3$ concentrations might reflect more direct signal of the NH$_3$ emissions than the NH$_3$ column density. If the ambient NH$_3$ measurements together with the satellite observations are used to inverse the NH$_3$ emissions, we would obtain more reasonable optimized emission estimates.

**Response**

We agree with the reviewer that near ground ambient NH$_3$ concentration observations might better constrain NH$_3$ emissions than the satellite NH$_3$ column densities. However, only 110 active sites are measuring bi-weekly average NH$_3$ concentration from the AMoN network in the U.S. The ground observations are too sparse to provide useful constraints in the inversion because of the high spatiotemporal heterogeneity of NH$_3$. Therefore, we decide to leave out the AMoN observations as an independent set of observations to evaluate the robustness of the inversion outcomes. It would be ideal if the two sets of observations can be used together in the inversion if more ground NH$_3$ observations become available in the future.

The sentence in line 395 is revised as follows: "*...shows that the optimized NH$_3$ emission estimates reduce the NMB between model outputs and independent observations, especially in April. The NRMSE remains high, indicating 1) the potential to further optimize NH$_3$ emission estimates when more representative observations of ambient NH$_3$ abundance becomes available; 2) the need to address the uncertainties in other processes affecting the NH$_3$ abundance, such as gas-particle partitioning, dry and wet deposition, and in cloud processes.*"
* * *
**Comment 5**

Technical comments

1. In lines 434-436 and lines 541-542: Please add the journals which the references are submitted to.

**Response**

The two references are updated as follows.

lines 434 – 436:

Cao, H., Henze, D. K., Shephard, M. W., Dammers, E., Cady-Pereira, K., Alvarado, M., Lonsdale, C., Luo, G., Yu, F., Zhu, L., Danielson, C. G., and Edgerton, E. S.: Inverse modeling of NH$_3$ sources using CrIS remote sensing measurements, Environ Res Lett, 15, 104082, 10.1088/1748-9326/abb5cc, 2020.

lines 541 – 542:

[revised manuscript text omitted]